

# Retrieval of ice water path from the FY-3B MWHS polarimetric measurements based on deep neural network

Wenyu Wang[1], Zhenzhan Wang[1], Qiurui He[1, 2], Lanjie Zhang[3]

[1]Key Laboratory of Microwave Remote Sensing, National Space Science Center, Chinese Academy of Sciences, Beijing 100190, China
[2]School of Information Technology, Luoyang Normal University, Luoyang 471934, China
[3]School of Information& Communication Engineering Beijing Information Science And Technology University, Beijing 100101, China

*Correspondence to*: Zhenzhan Wang (wangzhenzhan@mirslab.cn)

**Abstract.** Ice water path (IWP) is an important cloud parameter in atmospheric radiation, and there are still great difficulties in retrieval. The artificial neural network is a popular method in atmospheric remote sensing in recent years. This study presents a global IWP retrieval based on deep neural networks using the measurements from Microwave Humidity Sounder (MWHS) onboard the FengYun-3B (FY-3B) satellite. Since FY-3B/MWHS has quasi-polarization channels at 150 GHz, the effect of polarimetric radiance difference (PD) is also investigated. A retrieval database is established using collocations between MWHS and CloudSat 2C-ICE. Then two types of networks are trained for cloud scene filtering and IWP retrieval, respectively. For the cloud filtering network, using IWP of 10 g/m$^2$ and 100 g/m$^2$ as the threshold show the filtering accuracy of 86.48% and 94.22% respectively. For the IWP retrieval network, different training input combinations of auxiliary information and channels are compared. The results show that the MWHS IWP retrieval performs well at IWP >100 g/m$^2$. The mean and median relative errors are 72.02% and 46.29% compared to the 2C-ICE IWP. PD shows an important impact when IWP is larger than 1000 g/m$^2$. At last, two tropical cyclone cases are chosen to test the performance of the networks, the results show a good agreement with the characteristics of the brightness temperature observed by the satellite. The monthly MWHS IWP shows a good consistency compared to the ERA5 and 2C-ICE while it is lower than MODIS IWP.

## 1 Introduction

Ice clouds play an important role in the global climate (Liou, 1986), and their distribution strongly affects precipitation and water cycle (Eliasson et al., 2011; Field and Heymsfield, 2015). The long time series and global observation of ice clouds are essential for understanding the Earth's climate system. Depending on observation wavelength, satellite remote sensing can measure different cloud microphysics. Microwave measurement can penetrate deeper into cloud layers to measure thick and dense ice clouds, while infrared and visible instruments are mainly used for thin clouds measurement around the cloud-top (Liu and Curry, 1998; Weng and Grody, 2000; Stubenrauch et al., 2013). Since the measurements depend on microphysical properties of cloud particles (shape, size, concentrations), the individual instrument can only be sensitive to partly





information of clouds. Although ice water path (IWP) obtained from different instruments show several folds of differences (Stephens and Kummerow, 2007; Wu et al., 2009), it is of great importance to use remote sensing to get microphysical of clouds. Active observations such as lidar and radar as well as passive measurements such as visible/infrared imaging spectrometer and microwave radiometer have been used to get the cloud products (King et al., 1998.; Austin et al., 2009;

Delanoë and Hogan, 2010; Deng et al., 2010; Boukabara et al., 2011). Radiometers of millimeter frequencies are sensitive to larger precipitating hydrometeors while sub-millimeter frequencies are sensitive to smaller ice particles (Buehler et al., 2007). Cloud radar has the advantage of high vertical resolution and sensitivity than passive radiometer and can determine the vertical structure of ice clouds. However, this usually comes at the cost of low spectral range and spatial coverage of the observations (Pfreundschuh et al., 2020).

The brightness temperature (TB) depression caused by the scattering of ice particles is usually proportional to the IWP which simplifies the retrieval method from radiometric measurements (Liu and Curry, 2000). Researches about ice cloud retrieval using radiometers such as AMSU, SSMIS, MHS and MWHS, as well as limb sounders such as MLS, SMR, SMILES have been published for years (Zhao and Weng, 2002; Eriksson et al., 2007; Wu et al., 2008; Sun and Weng, 2012; Millán et al., 2013; He and Zhang, 2016). However, these spaceborne radiometers lack the ability of polarization

measurement while the dual-polarization measurements above 100 GHz show obvious polarized scattering signals of ice clouds. The recent theoretical model research indicates that the non-spherical and oriented ice particles are the main reason for the polarization signal (Brath et al., 2020).

With the increase of frequency, the polarimetric measurement will lead to a new understanding of clouds and their microphysical (Buehler et al., 2012; Eriksson et al., 2018; Coy et al., 2020; Fox, 2020). Most passive microwave sensors

which have dual-polarization channels are limited to frequencies below 100 GHz. However, these sensors are strongly affected by surface contamination. Currently, only GMI and MADRAS observed the ice cloud polarimetric signals above 100 GHz (Defer et al., 2014; Gong and Wu, 2017). By analyzing the polarization differences between the 89 GHz and 166 GHz channels of GMI, Gong and Wu (2017) found that large polarization occurs mainly near the convective outflow regions (anvil or stratified precipitation), while in the inner deep convective core and the distant cirrus regions, the polarization

signal is smaller. It is roughly estimated that neglecting the polarimetric signal in the IWP retrieval will lead to errors of up to 30% (Gong et al., 2018). Their further study showed that the main source of the 166 GHz high polarimetric radiance difference (PD) is horizontally oriented snow aggregates or large snow particles, while the low polarization signal could be small cloud ice, randomly oriented snow aggregates, foggy snow, or supercooled water (Gong et al., 2020). The Ice Cloud Imager (ICI) will provide a more comprehensive observation of ice clouds. By covering 100 GHz to 800 GHz, ICI has good

sensitivity to both large and small ice particles, and its dual-polarization channels also provide a way to observe horizontally particles (Eriksson et al., 2020).

The Microwave Humidity Sounder (MWHS) onboard the Fengyun-3B (FY-3B) satellite has been proven to give information about IWP. The MWHS has quasi-polarization channels at 150 GHz that can provide polarization information of





cloud ice, and it was hardly analyzed in past studies. The neural network is an easy way to find the nonlinear relationships
between TBs and IWP while the only problem is the lack of true values of IWP. CloudSat is recognized as a relatively
accurate instrument for cloud measurement, and its official Level-2C product is used in this paper. Many studies have been
performed to compare CloudSat products with in-situ measurements, the results show that the Level-2C product is quite
reliable when using a combination of Cloud Profile Radar (CPR) and Lidar. Its ice cloud water content (IWC) is quite close
to the in-situ observation (Deng et al., 2013; Heymsfield et al., 2017). Although CloudSat products still have considerable
uncertainties (Duncan and Eriksson, 2018), they can give us a relative accurate reference of IWP and IWC. Holl et al. (2010,
2014) present an IWP product (SPARE-ICE) that uses collocations between MHS, AVHRR, and CloudSat to train a pair of
artificial neural networks. The 89 GHz and 150 GHz channels were excluded since they are surface sensitive. However, the
150 GHz channel shows good sensitivity to precipitation-sized ice particles (Bennartz and Bauer, 2003). Brath et al. (2018)
retrieve IWP from airborne radiometers of ISMAR and MARSS using neural networks.

In this paper, we present an analysis of IWP retrieval from the FY-3B/MWHS observations based on the deep neural
network. The two 150 GHz channels and their PD are investigated. Firstly, we collocate the MWHS measurements with
CloudSat/2C-ICE IWP according to the observation time and geolocation. Secondly, we train deep neural networks (DNNs)
which are used to filter cloud scenes and retrieve IWP. The effects of different channels (include PD) and auxiliary
information on the DNN retrieval are also discussed. Finally, the performance of each network is evaluated and the
corresponding error is estimated. The trained neural networks are used for the IWP retrieval of two tropical cyclones and the
monthly averaged IWP map is compared with Aqua/MODIS L3 product and ERA5 reanalysis data. The main aim of this
study is to analyze the ability of the MWHS in IWP retrieval, especially the role played by the dual-polarization channels in
IWP retrieval.

This paper is organized to describe the data analysis in Sect. 2, followed by the retrieval method in Sect. 3. The IWP
retrieval results and analysis are discussed in the subsequent section, with conclusions in the end.

## 2 Satellite Observations

### 2.1 Instruments

#### 2.1.1 FY-3B/MWHS

The FY-3B satellite was launched on 5 November 2010, and MWHS was equipped as one of the main payloads. The MWHS
performs the cross-track scanning along the orbit with the angle of ±53.35° from the nadir to make 98 nominal
measurements per scan line, which is corresponding to the scanning swath of 2645 km in 2.667 s with a resolution of 15 km
at nadir. It measures at frequencies from 150 GHz to 190 GHz (two window channels at 150 GHz and three channels near at
183 GHz water vapor absorption line), these channels are labeled as CH.1 to CH.5 hereafter. The details of each channel are
listed in Table 1 (Wang et al., 2013). Compared to its successors (i.e. MWHS-II) onboard the FY-3C/D/E satellite, the 150





GHz channels of MWHS have quasi-horizontal and quasi-vertical polarization which may include unique information about clouds. These channels can give information near the Earth's surface and lower atmosphere, and can also be used to measure atmospheric cloud parameters. For the 150 GHz channels, Zou et al. (2014) investigated the polarization information and concluded that the polarization signal is related to the scanning angle and also to information such as surface wind speed, wind direction and salinity, especially in the clear-sky condition. The three channels around the 183.31GHz absorption line
aim to get profiles of atmospheric humidity. In all weather conditions except heavy precipitation, the five channels of MWHS can observe water vapor and ice in the atmosphere. In this study, the Level-1B brightness temperature dataset of MWHS is used.

Table 1. Channel characteristics of MWHS

| Channel | Central frequency (GHz) | Polarization | Bandwidth (MHz) | NEDT (K) |
|---|---|---|---|---|
| 1 | 150 | H | 1000 | 0.8 |
| 2 | 150 | V | 1000 | 0.8 |
| 3 | 183.31±1 | H | 500 | 0.9 |
| 4 | 183.31±3 | H | 1000 | 0.5 |
| 5 | 183.31±7 | H | 2000 | 0.5 |

**2.1.2 CloudSat/CALIPSO**

CloudSat is a cloud observation satellite launched into the NASA A-Train in April 2006, with a 94 GHz cloud profiling radar providing continuous cloud profile information (Stephens et al., 2008). The footprint size of CPR observation is about 1.3 km × 1.7 km, with a vertical resolution of 240 m. The scan time for each profile is about 0.16 s, and its sensitivity is -30 dBZ. It has an orbital inclination of 98.26°, which is similar to the FY-3B satellite. The Cloud-Aerosol Lidar and Infrared Pathfinder Satellite Observation (CALIPSO) was launched with the CloudSat satellite and designed to fly close to each other
in the A-Train satellite constellation to make synergistic observations. The Cloud-Aerosol Lidar with Orthogonal Polarization (CALIOP) carried on the CALIPSO is a dual-wavelength polarized lidar, providing 532 nm and 1064 nm backscatter profiles with a footprint of 75 m cross-track and 1 km along-track (Winker et al., 2009).

The CloudSat and CALIPSO Ice Cloud Characterization product (2C-ICE) contains retrieved estimates of IWC, effective radius and extinction coefficient for identified ice clouds measured by CPR and CALIOP with orthogonal polarization. The
2C-ICE cloud product uses a combined input of the radar reflectivity factor measured by the CPR and the attenuated backscatter coefficient measured by the Lidar at 532 nm to constrain the ice cloud retrieval more tightly than using only the radar product and to produce more accurate results (Mace and Deng, 2019). The combination of CPR and CALIOP provides a more complete measurement of the ice clouds than any other current spaceborne sensor measurements. Further study





showed that this combined retrieval method is less sensitive to the changes in the assumed microphysical properties than

CPR or CALIOP single retrieval (Delanoë and Hogan, 2010).

The 2C-ICE retrieval relies on forward model assumptions. Lidar is sensitive to small particles near the top of the cloud, but cannot measure that deep in the cloud which can lead to an unquantifiable error (Mace et al., 2009). A sensitivity study shows that multiple scattering, assumptions regarding particle habits and size distribution shapes are critical to the accuracy of the retrieval (Deng et al., 2010). The research also finds that the ratio between IWC product and in-situ measurements is

similar to the ratio between two independent in-situ measurements (around a factor of 2) and conclude that the retrieval agrees well with in-situ data. Since 2C-ICE is used to train the retrieval network in this work, the trained network directly inherits all the systematic errors and limitations of the product.

## 2.2 Collocation

Collocated measurement is the occurrence where two or more sensors observe the same regions at the same time. One factor

for the collocation window requirements is the specific observation target. Ice clouds is a fast-changing (minutes to hours) atmospheric parameter which needs a window with short time and small space. Another considered factor in defining the collocation window is the number of meaningful statistics for training.

The ascending node time of CloudSat is between 13:30 and 13:45 at the local solar time (LST) which is close to that of FY-3B (13:30 LST). Because of the close orbits and the ascending time between FY-3B and CloudSat, the number of

collocated measurements is quite large. In this study, a collocation dataset of MWHS and 2C-ICE was created by setting the collocation window to 15 min in time and 15 km in space. Since the footprint of MWHS is an order of magnitude larger than that of CPR, multiple 2C-ICE pixels are found within one MWHS measurement. Thus, the IWP values of 2C-ICE within a circular window (with a radius of 7.5 km) were averaged to represent the mean IWP for the MWHS measurement pixel. According to this collocation strategy, 1207731 collocations have been found between the FY-3B/MWHS and the

CloudSat/2C-ICE for the year 2014. Since the different observation methods of MWHS and CPR/CALIOP, only 14 2C-ICE pixels are contained in the best case of collocations (See Fig. 1a). Thus, the CloudSat footprints cover at most 13.75% of the area of an MWHS footprint, an error from imprecise collocation is unavoidable and the representation of the dataset must be considered. The criteria discussed in Holl et al. (2010) is applied to reduce the sampling effect of collocations. However, in the case of highly inhomogeneous clouds, larger uncertainty for the IWP within MWHS pixels cannot be eliminated.

Figure 1 illustrates the statistics of 2C-ICE IWP within the MWHS footprints in the collocations. In most cases, more than 10 IWP pixels of 2C-ICE were averaged in the corresponding MWHS pixel. However, there are still many MWHS pixels that only cover a small quantity of 2C-ICE pixels which means the collocations are poorly represented. The coefficient of variation of each collocation pixel is manifested in Fig. 1b. The coefficient of variation is used to represent the IWP dispersion of 2C-ICE pixels in each MWHS pixel. When the coefficient of variation is small, it means the IWP of 2C-ICE





pixels averaged in this MWHS pixel are homogeneous and represent the scene that MWHS observed relatively well. Overall, the collocation dataset is available for subsequent processing.

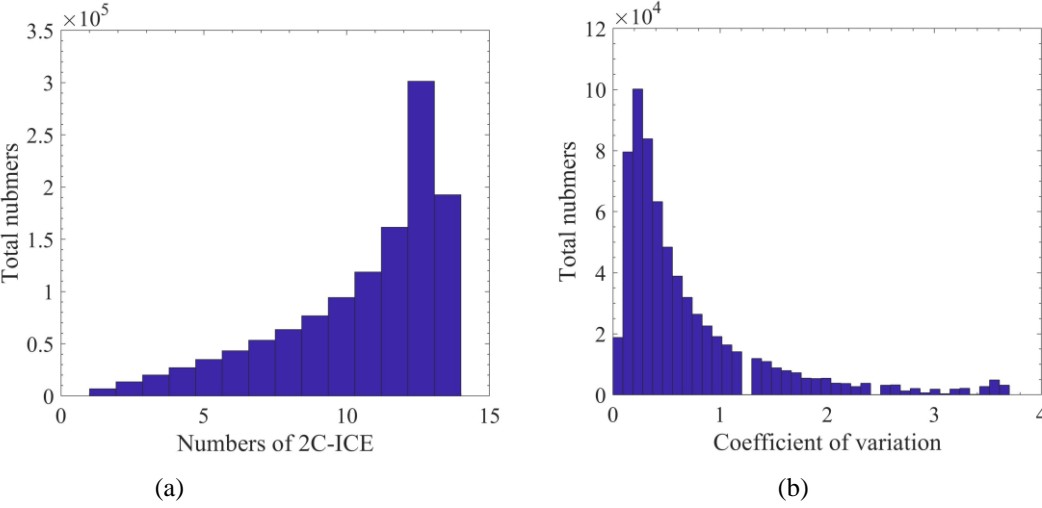

Figure 1. Statistical information of MWHS and 2C-ICE collocations in 2014. (a) Histogram of the number of 2C-ICE pixels within an MWHS pixel. (b) Histogram of the coefficient of variation of the collocations.

Figure 2 and Figure 3 give statistical information on the scan angle, latitude and time of the MWHS measurements in this dataset. Since the dataset is used for global retrieval, it must have sufficient samples and their distribution must represent the real world. According to the statistical results of the collocated MWHS pixels shown in Fig. 2, most collocation occurred in one side of the flight direction (from the 40th to 90th scan pixel). To the observation latitude, the collocations near the nadir scan (the 49th pixel) cover the latitude from 80°S to 80°N, while at the edge of the observation (the 90th pixel) they only
cover the tropical regions. In terms of observation time and latitude, Figure 3 illustrates that there is an obvious lack of data above 60°S from April to September, and there are also few data between 0° and 30°S in December. The data distribution suggests that the training in polar regions may be inadequate.





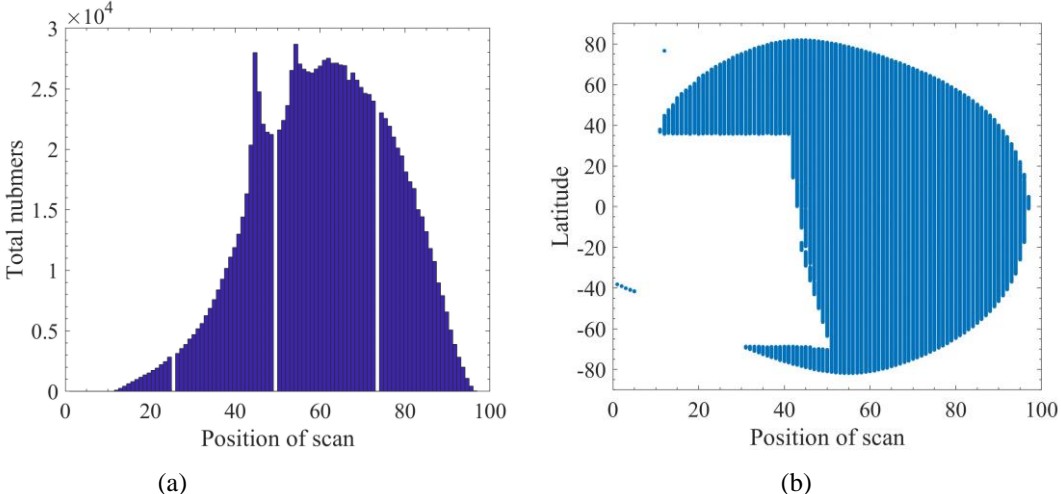

(a)                                                    (b)

Figure 2. Statistical information of scan angle and latitude of MWHS observations in the collocation dataset.

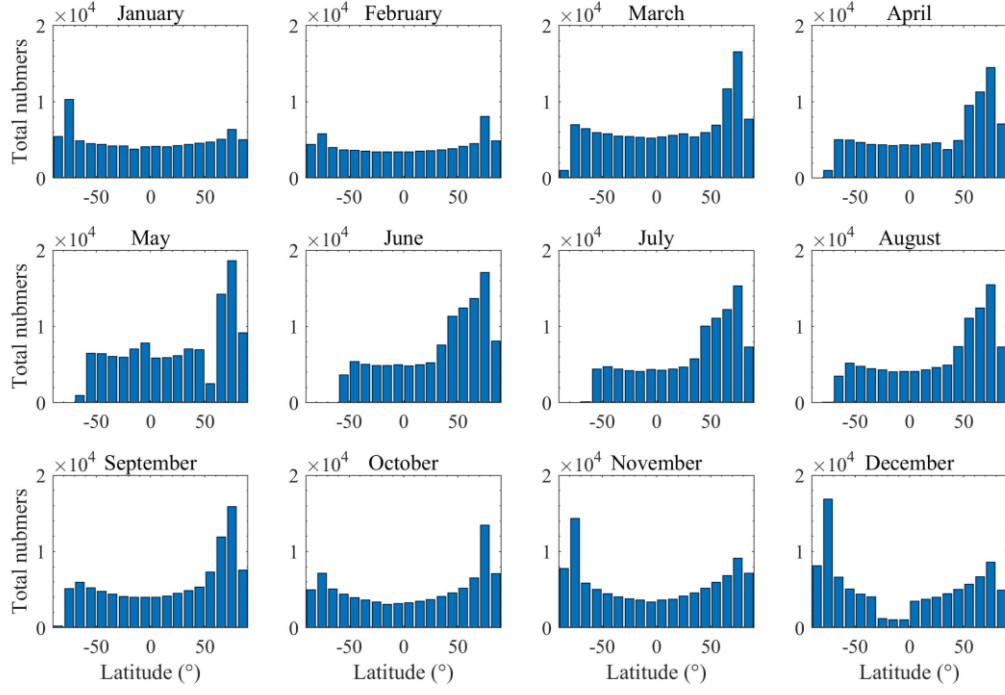


Figure 3. MWHS measurements distribution of time and latitude in the collocation dataset.

For IWP retrieval, the collocations should be classified into two bins (clear-sky scene and cloudy scene) according to a specific IWP threshold. A threshold of IWP >10 g/m2 is preliminarily selected to classify cloudy scenes. Thus, 401520 collocations are recognized to be cloudy scenes in this dataset.





Figure 4 and Figure 5 show the statistics of PD and TB (clear-sky and cloudy) at 150 GHz over the ocean and the land in
2014. The scatter points are color-coded by IWP from 2C-ICE. There is a clear difference in the shape of the scatter plot
between the ocean and the land. There is a significant peak of PD at TB of 210 K to 250 K over the ocean while there is no
peak of PD over the land. This may be related to the strong polarization signal at the ocean surface. Compared the cloudy
scene with the clear-sky scene, the scatter shapes are quite similar over the ocean and the land except when IWP is larger
than 1000 g/m2. The PD (around 5 K) in both cloudy and clear-sky may be caused by the scan angle due to the quasi-
polarization measurement of MWHS. It also can be found that the PD which may be caused by ice particles is about 5 K
when IWP is larger than 1000 g/m2. This is consistent with the retrieval results shown in section 4.2.

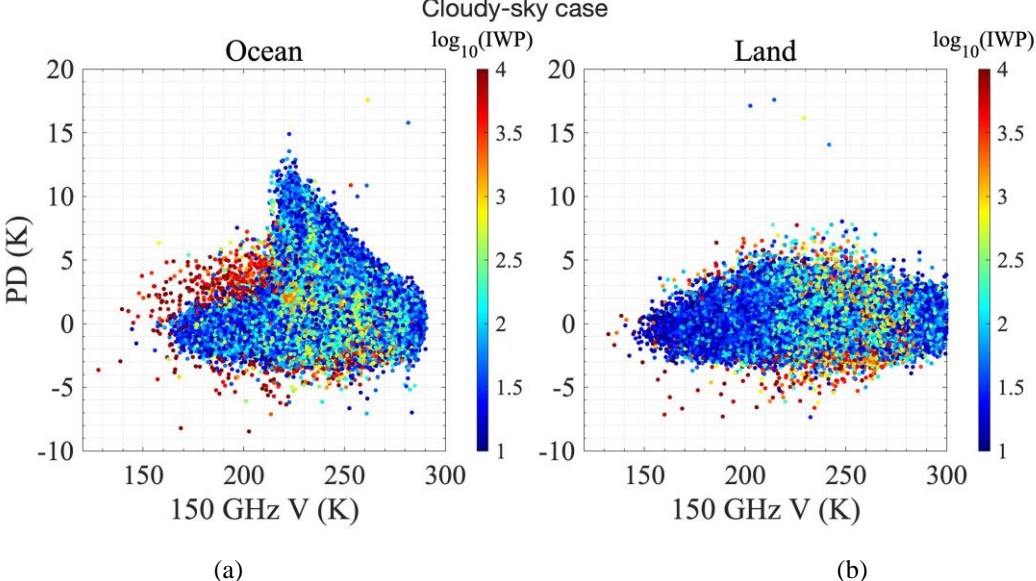

Figure 4. The PD–TB150V scatter plots for the collocations in the cloudy scenes over the ocean (a) and land (b). The points
are color-coded by IWP from 2C-ICE.





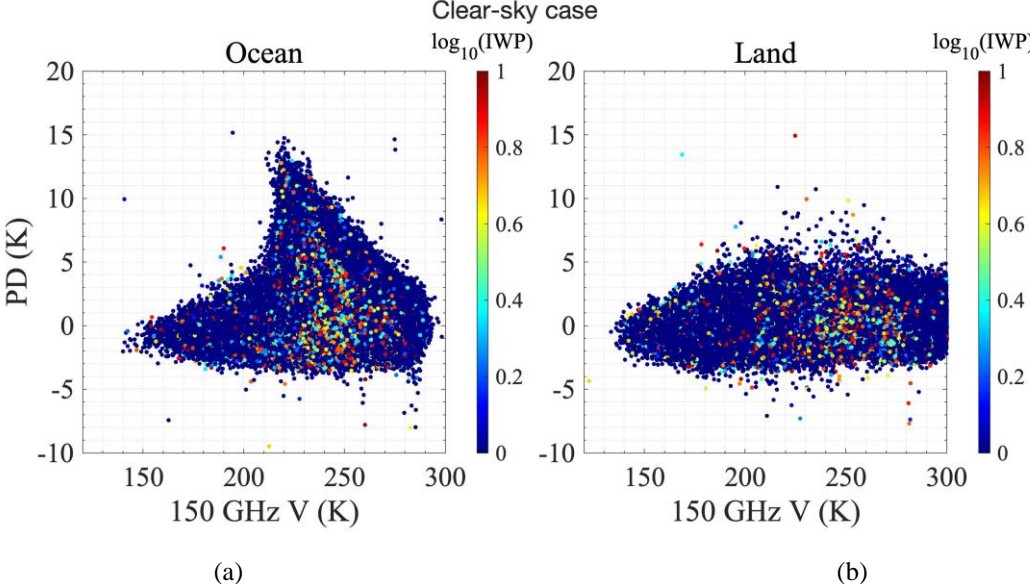

(a)              (b)

Figure 5. Same as Fig. 4 but for clear-sky scenes.

## 3 Retrieval method

The collocations are used as a retrieval database to train the networks, the processing flow is shown in Fig. 6. The DNN is

185 a feed-forward neural network which contains an input layer, several hidden layers, and an output layer. The DNN is a fully connected network, neurons in each layer connect with all neurons in the next layer. The hidden layers are used to perform the nonlinear calculation to achieve a nonlinear mapping of the relationship between input and output data. DNN is based on backpropagation learning algorithms to search for a minimum loss function (such as the mean squared error between prediction data and reference data) and then adjust the thresholds and weights iteratively to close the reference data. The

190 outstanding nonlinear mapping capability makes DNN popular for geophysical retrieval.

In this study, DNN with 6 layers is selected. The first layer is the input layer, and each input quantity uses a neuron to connect with the next layer. The second to fifth layers are the hidden layers, in which 300 neurons are used for each layer, and the Rectified Linear Unit (ReLU) is selected as the activation function. Since networks are prone to overfitting in the training, the early stopping method is used to improve the training. To remove the effect of the order of data, random

195 assignation and normalization are performed in the front of the hidden layers. The final layer is the output layer which uses the IWP of 2C-ICE (transfer to log space) as reference. The activation function of the last layer is selected according to the target of the network. For the determination of cloudy and clear-sky scenes, the sigmoid function is used for binary classification. For the IWP retrieval, the results are output directly. In the iterative training of the networks, the models with the best results in the validation data will be retained. The hyperparameters were chosen by comparing the performance of



DNNs with different hidden layers, number of hidden neurons and regularization parameters. Each network mentioned in the next section uses the same hyperparameters of the model to ensure that the performance of the network is only affected by the input parameters.

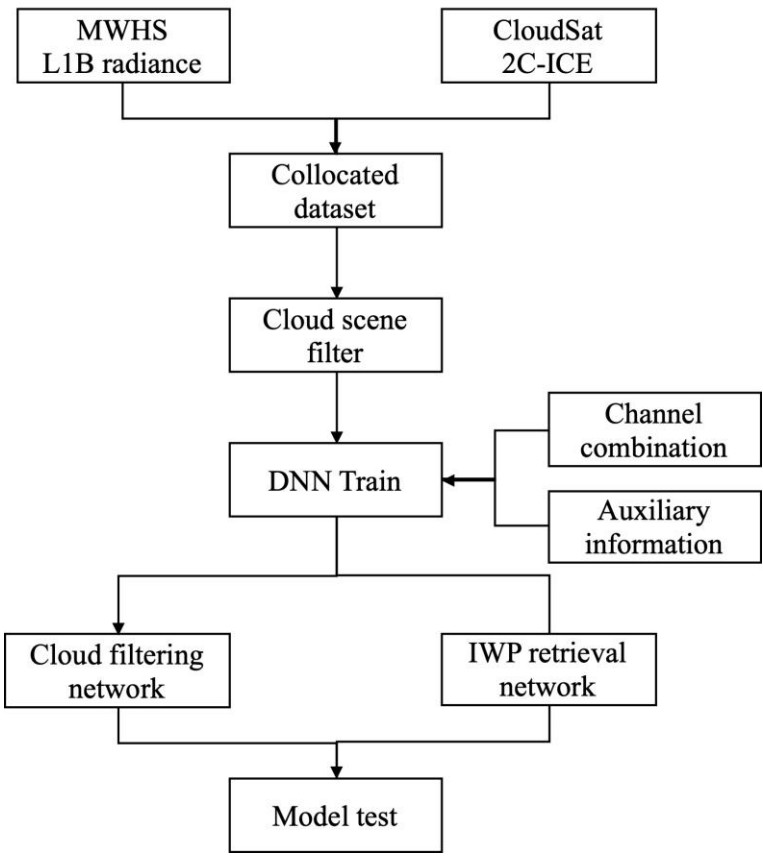

Figure 6. The schematic of the MWHS retrieval based on the DNN model.

The sensitivity of ice cloud is discussed by Holl et al. (2010) and Eliasson et al. (2013), their studies show that no significant radiance signals at IWP <100 g/m$^2$ for MHS measurements. Thus, two thresholds for cloud filtering are considered respectively with 2C-ICE averaged IWP of 10 g/m$^2$ and 100 g/m$^2$ within the MWHS footprint. For the IWP threshold of 10 g/m$^2$, 401520 collocations are left. For the IWP threshold of 100 g/m$^2$, 168898 collocations are left.

From those collocations, we randomly assign 75% to be used for training and 25% to be used for validation. The training
data are used as a sample of data for model fitting. The validation data can be used to tune the hyperparameters of the network and for preliminary evaluation of the model. Collocations during January 2015 are used for testing. These data are not used to train the networks and adjust the hyperparameters but serve as independent data to test the performance of the final obtained networks.





## 4 Results

To retrieve IWP from MWHS measurements, two networks are trained for different capabilities. The first one is to classify a scene whether is clear-sky or cloudy. The other one is to retrieve IWP. The two networks are used separately, and the IWP of the scene which is considered to be clear-sky is set to 0.

### 4.1 Cloud Filtering Network

The network structure, training dataset and cloud IWP thresholds are discussed above. The sigmoid activation function can
make the output of the network range from 0 to 1 which represents the probability of the cloud occurs. Thus, a threshold value of cloud probability must be assigned to determine the cloudy scene. After testing, a threshold value of 0.5 is the most appropriate for cloud filtering. The accuracy is 86.48% for IWP threshold $>10$ $g/m^2$ and 94.22% for IWP threshold $>100$ $g/m^2$. Thus, IWP $>100$ $g/m^2$ is used as the threshold for cloud filtering.

### 4.2 IWP Retrieval Network

For the global IWP retrieval, clear-sky scenes are excluded from the training dataset. Different combinations of the network input are compared to find the best retrieval strategy. These cases are divided into three parts, their details and mean retrieval errors are listed in Table 2. To avoid the effect of random factors, each combination was run twenty times and the best result was taken.

   The first part (Case 1-5) considers the distribution of the training data. Collocations selected from IWP $>10$ $g/m^2$ and
IWP $>100$ $g/m^2$ are considered separately (Case 1 and Case 2). The results show that Case 1 is not converged well in the training. Its mean error is much larger than that of other cases. Thus, the IWP threshold of 100 $g/m^2$ is used in the following cases. It is consistent with the results of cloud filtering. There are 47976 collocations over land and 112085 collocations over the ocean in this dataset. Data over land and ocean are also retrieved separately (Case 3 and Case 4). Case 5 uses the better representative collocations (i.e. sample optimization discussed in section 2.2) to train the network. In this case, MWHS pixel
with more than 10 pixels of 2C-ICE and less than 0.6 coefficients of variation are selected which lead to the collocations reducing to 100249. Observations over the ocean show a mean relative error of 81.77% while retrieval over land shows a poorer result of 100.92%. This may be due to the insufficient amount of data over the land. Applying the criteria of sample optimization, the result is improved compared with Case 2 which means the collocations are sampled well. Case 5 have the mean and median relative error of 96.72% and 52.84% respectively and it is used in Part II as the basic input. Figure 7 shows
the mean and median relative errors of Case 2-5 using MWHS all channels with reference IWP divided into several bins from 100 $g/m^2$ to 10000 $g/m^2$. The performance of the retrieval is poor (mean errors $>150\%$) between IWP of 100 $g/m^2$ and 200 $g/m^2$ while the errors reduce to 45-70% at IWP $>200$ $g/m^2$. It should be noted that there are only 13 samples in the last bin (i.e. IWP $>10000$ $g/m^2$).





Table 2. Mean relative errors of IWP retrieval

| Part | Combination Case | Mean Error (%) | Median Error (%) |
|---|---|---|---|
| I. Data distribution | 1. IWP >10 g/m$^2$ | 170.17 | 68.60 |
| | 2. IWP >100 g/m$^2$ | 98.01 | 54.23 |
| | 3. Only Land | 100.92 | 57.91 |
| | 4. Only Ocean | 81.77 | 51.45 |
| | 5. Sample Optimize | 96.72 | 52.84 |
| II. Auxiliary information | 6. Latitude | 83.49 | 49.65 |
| | 7. Scan Angle | 101.16 | 52.74 |
| | 8. Mask | 80.42 | 50.41 |
| | 9. Latitude + Scan Angle | 81.23 | 49.86 |
| | 10. Latitude + Mask | 79.28 | 48.38 |
| | 11. Scan Angle + Mask | 86.27 | 52.15 |
| III. Channel selection | 12. CH.1-CH.5 | 72.02 | 46.29 |
| | 13. CH.2-CH.5+PD | 74.68 | 46.45 |
| | 14. CH.2-CH.5 | 73.20 | 47.91 |
| | 15. CH.3-CH.5+PD | 82.79 | 50.13 |
| | 16. CH.3-CH.5 | 82.21 | 50.99 |

The second part (Case 6-11) aims to select the important auxiliary information which is useful for the retrieval. Based on Case 5, additional information including latitude, scan angle and ocean/land mask is added to train the networks. In terms of the mean errors, a significant improvement of retrieval performance is achieved by adding latitude or ocean/land mask information and even more improvement when they are all added. In MWHS measurements, the signal from ice clouds is a reduction in TB through scattering effects. When lacking the latitude information, it is difficult to distinguish whether the
decrease of TB is due to the ice particles or the low radiance from the surface or atmosphere. So is the ocean/land mask information. Case 10 (Latitude + Mask) shows a mean error of 79.28% which is improved by 17.44% compared to Case 5. The scan angle seems to have a negative impact on the retrieval (Case 7). However, when it is combined with latitude or





mask information, the results are also improved (Case 9, 11). The retrieval error of each bin is shown in Fig. 8. Compared to the first part, adding auxiliary information can significantly reduce the retrieval errors, especially when the IWP is small.

The third part (Case 12-16) compares the performance of different channel combinations (all the auxiliary information is added). In this part, the influence of the 150 GHz channel and PD is mainly focused on. According to the retrieval errors, the water vapor channels (CH.3-CH.5) at 183 GHz show good sensitivity to IWP (Case 16) while adding the window channel (CH.2) give a large improvement to the retrieval (Case 14). It might be due to the sensitivity of 150 GHz to the deep convective clouds and precipitating ice. Case 12 and Case13 exhibit similar retrieval performance which means the network

can discriminate the polarization information well. Figure 9 illustrates the errors in different IWP bins which give a more detailed comparison. Compare Case 15 with Case 16, adding PD give an obvious improvement on the retrieval results at IWP >1000 g/m². The mean and median errors are reduced by ~10% and ~15% at IWP >2000 g/m², respectively. However, it increases a little mean and median error at IWP <500 g/m². It can be explained that PD at the window channel is also sensitive to the surface which will make the network misidentify the PD from the surface as IWP when the cloud is thin.

When the cloud is thick, more oriented ice particles appear to cause a relatively large PD. It can be improved by adding the surface-sensitive channel simultaneously. Case 13 illustrates that adding CH.2 significantly reduces the retrieval error at IWP <400 g/m² and maintains the benefit from adding PD at IWP >1000 g/m² (compare Case 13 and Case 15). Case 13 and Case 14 also demonstrate the improvement in retrieval performance by adding PD at IWP >1000 g/m². Figure 10 shows the scatter plot between MWHS IWP and 2C-ICE IWP in January 2015.

Since the 2C-ICE retrievals are proved to be mostly no more than a factor of 2 from the in-situ measurements (i.e. 100% error), it gives a lower bound of the network retrieval error. According to the law of error propagation, if both are random and uncorrelated, the true mean error of the best network retrieval (Case 13) can be calculated as $\sqrt{1^2 + 0.7202^2} \times 100\% = 123.24\%$ and the true median error can be calculated as $\sqrt{1^2 + 0.4629^2} \times 100\% = 110.10\%$.





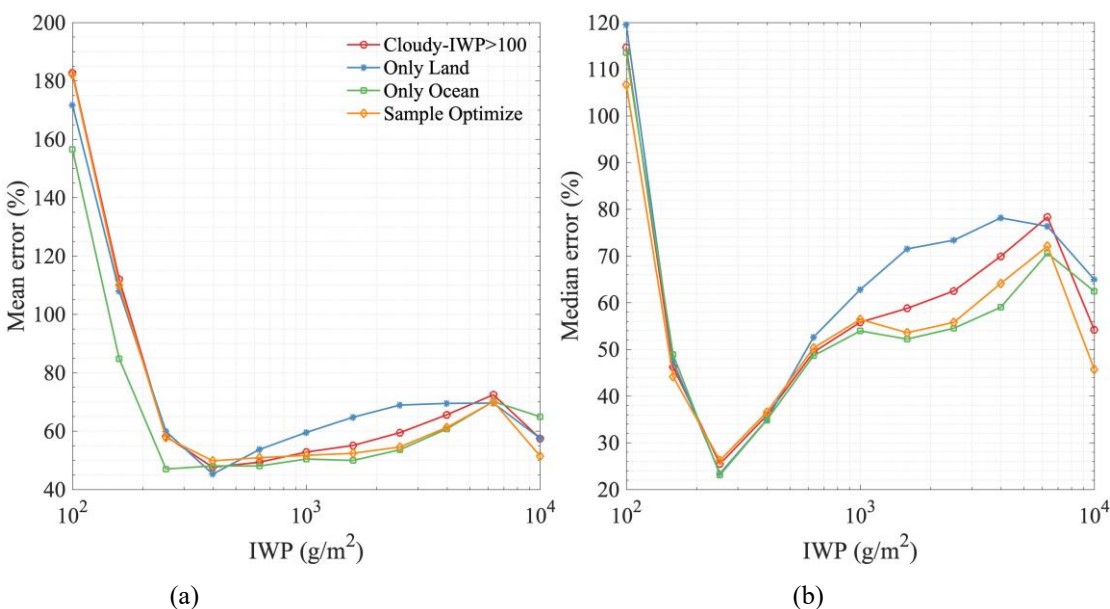

Figure 7. Performance for a global retrieval based on MWHS CH.1-CH.5, with different cloud scenes selected in the training.

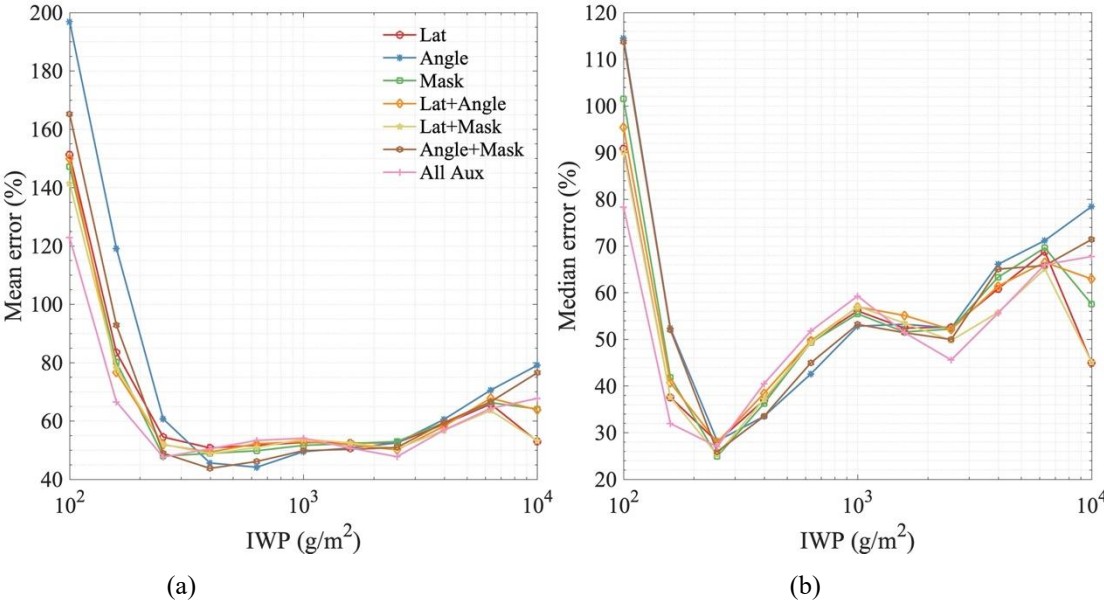

Figure 8. Comparison between the performance of the IWP retrieval networks using different auxiliary information combinations of input data. The "All Aux" in the legend is equivalent to Case 12 in Table 2.



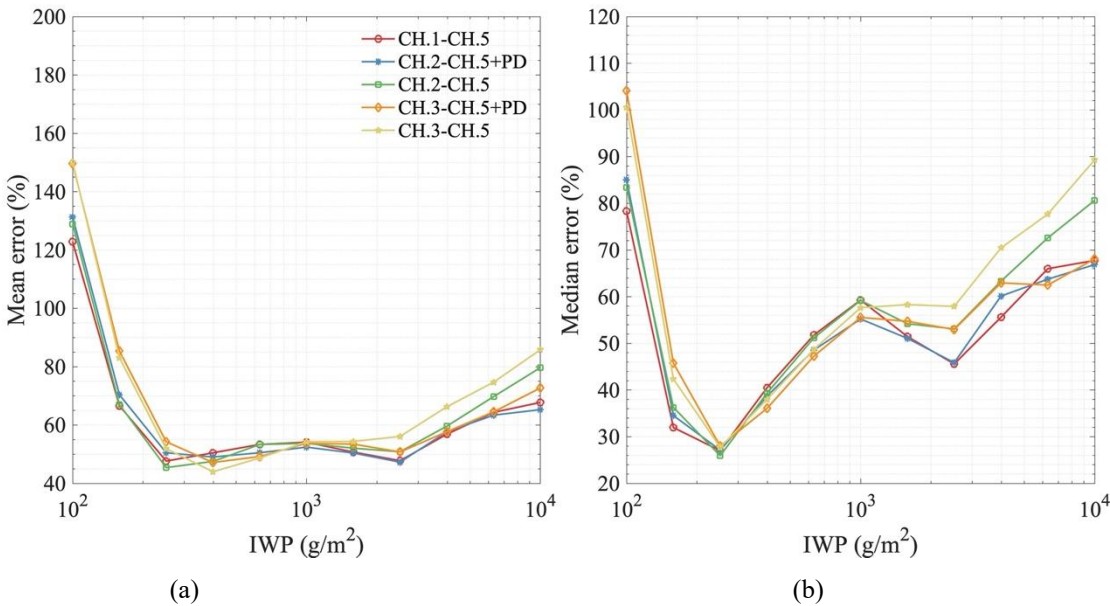

<p style="text-align:center">(a)                    (b)</p>

Figure 9. Comparison between the performance of the IWP retrieval networks using different channel combinations of input data. All the auxiliary information is added.

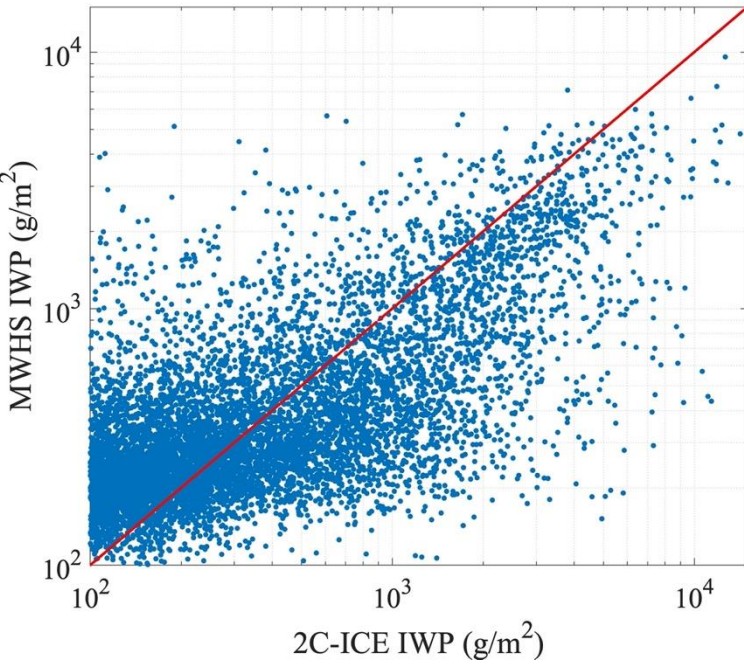

Figure 10. Comparison between 2C-ICE and MWHS IWP. The red line represents the diagonal 1:1 line. Clear-sky scenes are excluded.





### 4.3 Network application

#### 4.3.1 Tropical Cyclone IWP retrieval

MWHS observed TB in the lifetime of the typhoon Rammasun that occurred in July 2014 is manifested in Fig. 11. Three regions of typhoon (eye, eyewall, and spiral rain bands) is obvious. Quite low TB (as low as 150 K) can be found in the regions of the eyewall and spiral rain bands which is mainly caused by the scattering of ice particles in the clouds. There is no significant difference between the TB over the ocean and the land. According to the PD at 150 GHz ($TB_V - TB_H$), its distribution characteristics are the same as the typhoon. The PD reaches its maximum in the anvil precipitation regions (around 5 K, consistent with the result in Fig. 4) and decrease in the remote clear-sky or cirrus regions. All channels show similar typhoon characteristics.

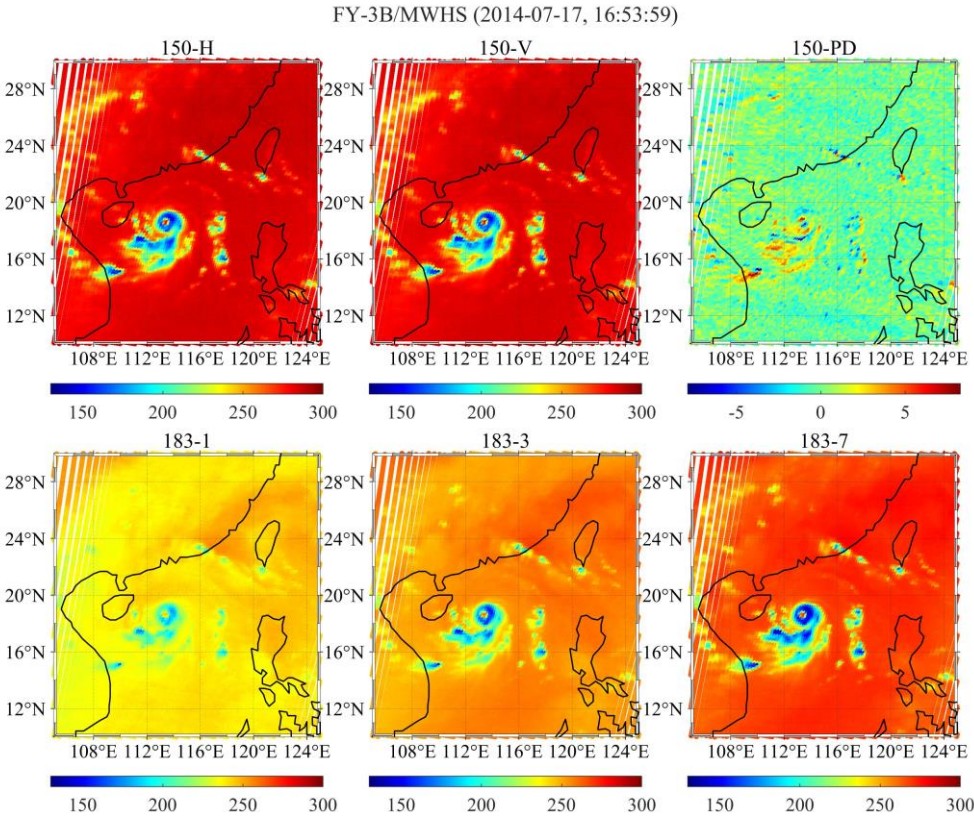

Figure 11. Typhoon Rammasun on 17 July 2014 as observed with FY-3B/MWHS channels. The polarimetric differences are shown separately.

Comparing the 37 GHz and 89 GHz TB of Microwave Radiometer Imager (MWRI) onboard the same satellite, the 89 GHz channels of MWRI show similar but higher TB in the typhoon regions which means the 89 GHz TB is not only sensitive to cloud ice but also contaminated by liquid water and ocean surface. The 37 GHz TB shows the complete opposite





characteristics to the 89 GHz TB since it is only sensitive to liquid water which greatly increases the TB. By comparison, the 37 GHz TB shows a significantly high temperature in the typhoon regions, the 150 GHz TB shows a significantly low temperature in the typhoon regions due to its sensitivity to ice particles, and the 89 GHz TB is lower than the 37 GHz but higher than the 150 GHz in the typhoon regions due to its sensitivity to both ice particles and liquid water. Thus, the surface

temperature and emissivity are quite important if the 89 GHz measurements are used in the retrieval.

The PD from 37 GHz channels shows a clear division between the land, ocean and typhoon which is opposite to the TB map. There is no polarization signal from the land and the polarization difference from the ocean is large. The PD in the typhoon region is almost zero because the 37 GHz TB is not affected by ice particles. And the polarization signal from the ocean surface cannot reach the sensor due to the influence of liquid water. The PD from 89 GHz measurements contain a

joint contribution from the ocean surface and ice particles which show about 0 K at land and 5-10 K in the typhoon regions and much larger at the ocean surface. There is no significant difference in the PD between the ocean (non-typhoon regions) and land which means the 150 GHz channel is not strongly affected by the surface polarization signal and can provide a good representation of the polarization information of cloud ice particles. From the comparison above, PD of 150 GHz in the typhoon region does come from the ice cloud rather than the ocean surface. The PD from MWRI 37 GHz, 89 GHz and

MWHS 150 GHz measurements in the lifetime of another typhoon Vongfong is shown in Fig. 13. The characteristics are similar to those in Fig. 11 and Fig. 12. The PD from the 150 GHz measurement shows consistent characteristics with the typhoon shape. Therefore, the PD of 150 GHz is believed to play a significant role in the retrieval of IWP.





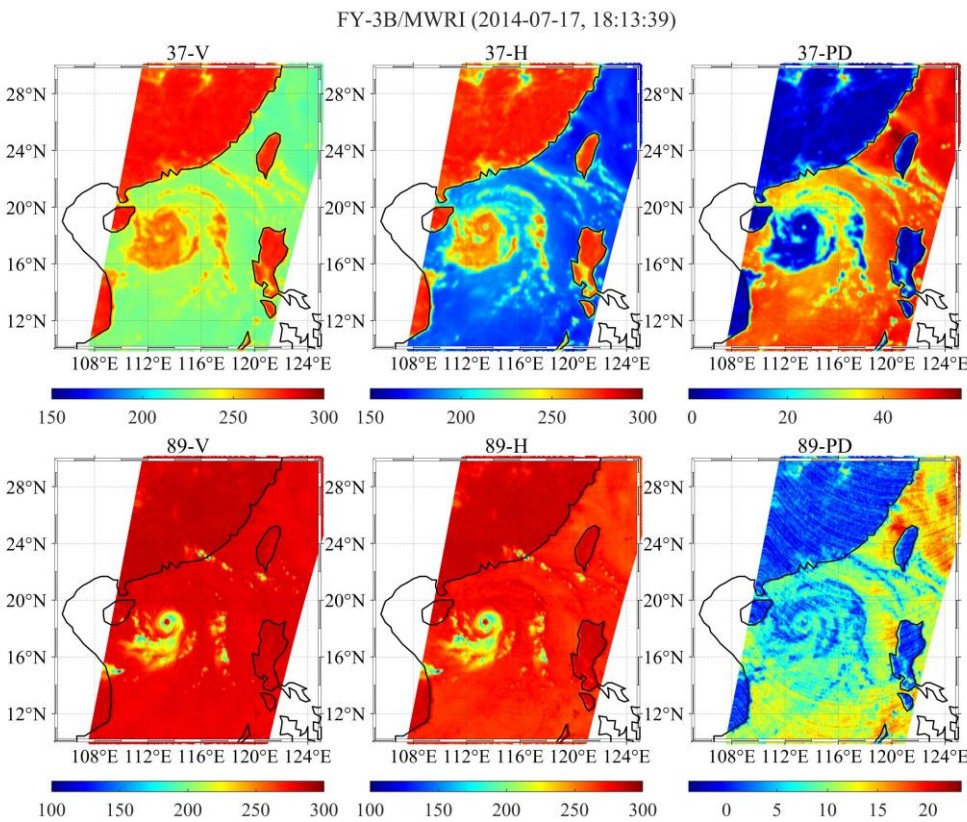

Figure 12. Typhoon Rammasun on 17 July 2014 as observed with FY-3B/MWRI 37 GHz and 89 GHz channels. The polarimetric differences are shown separately.

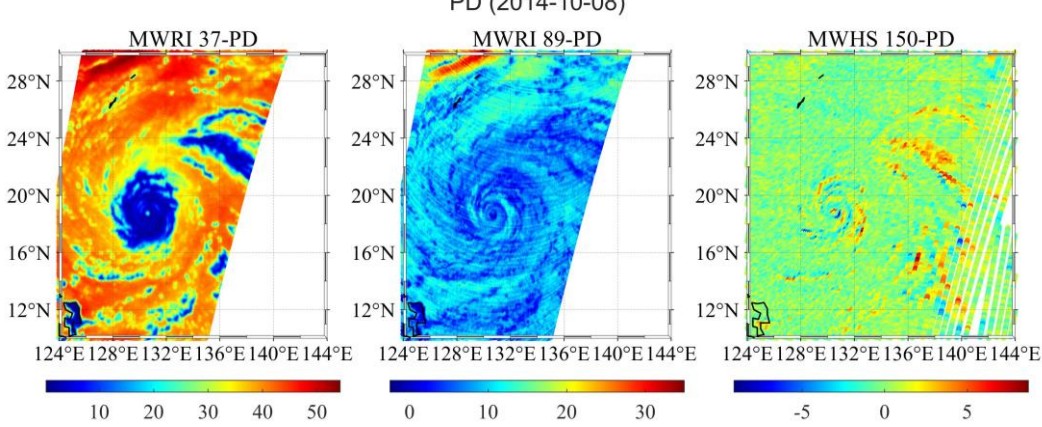

Figure 13. PD of MWRI and MWHS for Typhoon Vongfong on 8 October 2014.





Applying the two neural networks trained above to the two typhoon scenes, IWP maps of the two typhoons are retrieved, as shown in Fig. 14. In this figure, the structure and the distribution of IWP are consistent with the characteristics of TB and
PD showed in Fig. 11-13. The largest IWP (as large as 10000 g/m²) is found in the eyewall regions. The spiral rain bands also show an obvious IWP of 2000 g/m². There is no IWP in the eye of the two typhoons which is consistent with the actual. The white regions are the clear-sky which is filtered by the cloud filtering network. At this time and space scale, the performance of the two neural networks appears to be good.

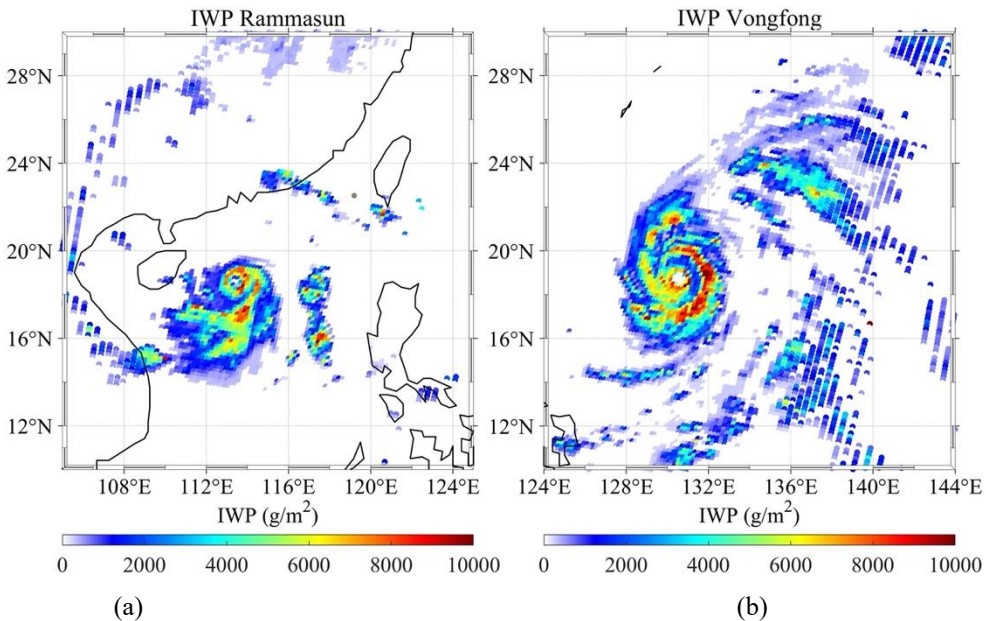

Figure 14. MWHS IWP retrieval results at two typhoon scenes shown in Fig. 11 and Fig. 13. The white regions are the clear-sky distinguished by the cloud filtering network.

### 4.3.2 Global mean IWP comparison

Figure 15 and Figure 16 show the monthly global mean IWP for summer (July 2014) and winter (January 2015) from Aqua/MODIS L3 product (MYD08_M3, Platnick et al., 2015), CloudSat 2C-ICE, FY-3B/MWHS retrieval and ERA5
reanalysis dataset. ERA5 IWP data shown here is combined of its total column snow water and cloud ice water data since it differentiates between precipitating and non-precipitating ice. Since 2C-ICE is used to train the networks, MWHS IWP is certainly approaching the 2C-ICE. However, due to the low cover of CloudSat, the monthly mean IWP of 2C-ICE with 5° grid still lack sufficient observations. The MWHS IWP is in general consistent with the range and trend of the ERA5 IWP while there are two obvious regions with large IWP found by MWHS in Tibetan Plateau and Hudson Bay, January 2015.
Compared to the MODIS IWP, MWHS IWP is lower in parts of the midlatitudes and polar areas. This may be due to the





different measurement resolutions or the bright surfaces which is covered by snow and ice. Because there is no independent in situ measurement, it is difficult to validate which one is more correct. From an overall perspective, it is indicated that the performance of the cloud filtering and IWP retrieval network is good.

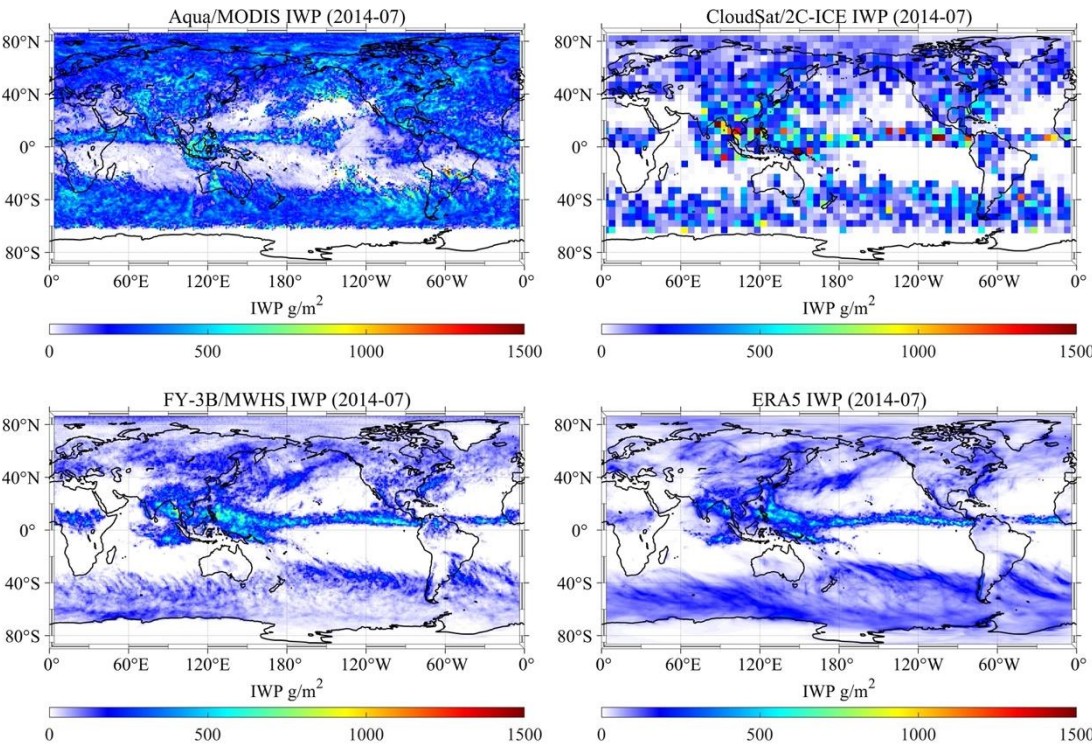

Figure 15. Monthly global mean IWP from MODIS, 2C-ICE, MWHS and ERA5. 2C-ICE is gridded on a 5° grid, while the other products are gridded on a 1° grid.



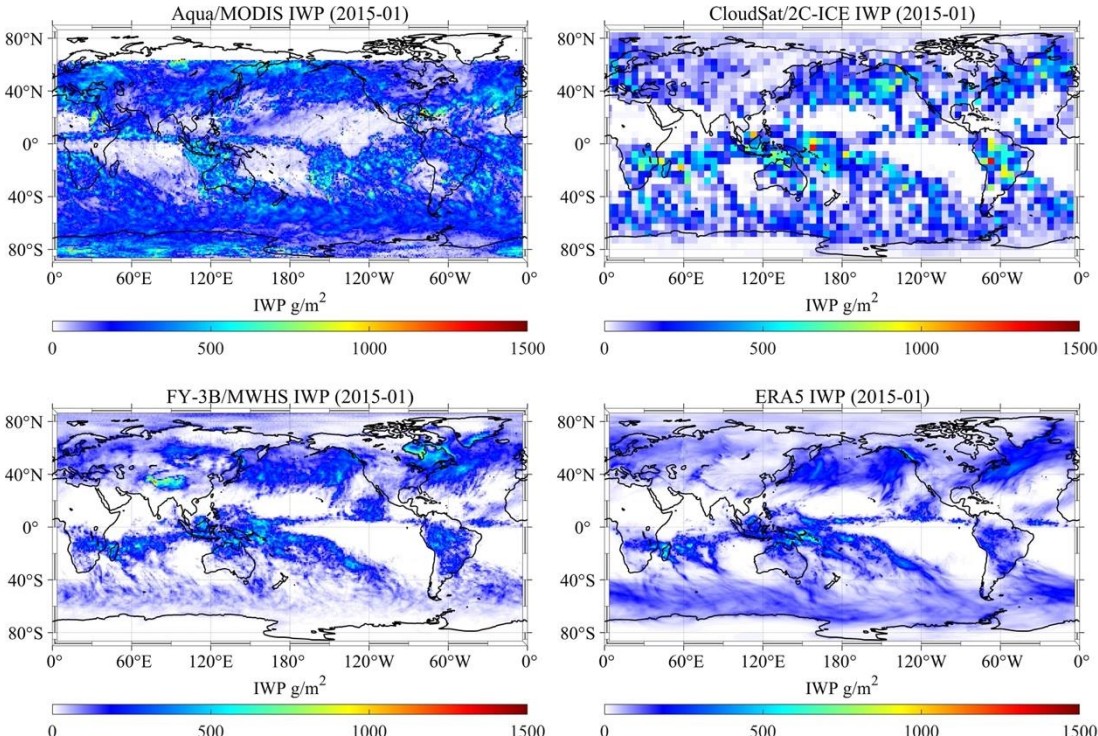

Figure 16. Same as Fig. 15 but for January 2015.

**4.4 Discussion**

From the error curves shown in Fig. 7-9, the trend is the same for almost all combinations: the errors are largest at IWP =100 g/m$^2$, then decrease rapidly and keep smooth, and finally, there is a small increase again at IWP >2000 g/m$^2$. The largest error (at IWP =100 g/m$^2$) is supposed to be caused by the insensitivity of the MWHS channels to the thin clouds. The increasing error at IWP >2000 g/m$^2$ could be due to the insufficient training of the network (fewer samples). Ice cloud misidentification is an unavoidable problem mainly due to the mismatch between CloudSat and MWHS footprints spatially

and temporarily. Since CloudSat pixels only cover less than 15% of the MWHS pixel, the 2C-ICE scenes cannot fully represent the MWHS observations, especially in the case of thin clouds. However, the results show that the neural network does well in IWP retrieval, with the mean difference from 2C-ICE at about 70%, which is acceptable for passive radiometer observation. The wide coverage and day/night availability of MWHS facilitates continuous and global IWP observation. PD at 150 GHz seems not to play a significant role in the overall retrieval errors, this is because the PD mainly occur at

IWP >1000 g/m$^2$, at which the error is smaller than the error at IWP =100 g/m$^2$ and the samples of IWP >1000 g/m$^2$ are also much smaller, so the effect of PD is greatly diluted in the overall average. In addition, the PD of quasi-polarization channels is related to the scan angle and does not fully represent the polarization information of the ice particles, especially near the 45° scan angle. However, from the error bins, PD does have a positive effect on retrieval.





However, there are also some difficulties of IWP retrieval using neural networks, including the representative of the
collocations, the requirement of a sufficient number of samples, the unexplainable of the network hyperparameters, etc. And
in the case of not being able to obtain a large number of true samples, the use of neural networks can only converge to a
certain observed product which has the highest accuracy. If the model simulation results (typical profile) are used for
training, the generalization ability of the network will strongly depend on the model itself and the input fields. The
microwave band below 200 GHz is only sensitive to large ice particles and thick clouds, adding the infrared channels like
SPARE-ICE does can improve the retrieval. Information about surface temperature and emissivity also seem important for
IWP retrieval.

## 5 Conclusions

In this paper, an analysis of global IWP retrieval from FY-3B/MWHS radiance measurements based on neural networks is
presented. MWHS onboard FY-3B satellite has two quasi-polarization channels at 150 GHz which was overlooked in
previous studies. For IWP retrieval, CloudSat/2C-ICE is chosen as the reference dataset for neural networks because it is
publicly available and it meets the requirements in terms of data numbers and measurement accuracy. The collocation
between MWHS and 2C-ICE is performed by using a window with 15 min in time and 15 km in space (a radius of 7.5 km).
Comparing the TB and PD at cloudy and clear-sky scenes, PD caused by ice particles can be easily identified at IWP >1000
$g/m^2$ with a typical value of 5 K. Two types of networks are trained to retrieve the IWP from MWHS measurements. Cloud
filtering network is trained to classify the cloudy and clear-sky scenes. For the IWP threshold of 10 $g/m^2$ and 100 $g/m^2$, the
accuracy of cloud filtering is 86.48% and 94.22% respectively. IWP retrieval networks with different combinations of
channels and auxiliary information as input are compared to find the best retrieval strategy. The retrieval results show that
the network performs well at IWP >100 $g/m^2$. The mean and median relative error of the best case is 72.02% and 46.29%
compared to 2C-ICE. Adding the 150 GHz channel gives an obvious improvement on IWP retrieval and the PD also make a
good impact when IWP is larger than 1000 $g/m^2$. The trained two networks show good performance in the application cases
of Typhoon Rammasun and Vongfong. The monthly mean IWP from MWHS is similar to that of CloudSat and ERA5,
which makes the retrieval results more credible. Compared to the IWP product from MODIS, the MWHS IWP is
significantly lower in the middle to high latitudes.

Neural networks are widely used to statistically characterize the mapping between radiometric measurements and related
geophysical variables. The advantages of neural networks are their simplicity and ease of use, their ability to effectively
learn the complex nonlinear mapping relationships in samples, and their better robustness to noisy data. By using the
collocated measurements, there is no need to establish a complicated radiative transfer model with many possible sources of
error. Although the retrieval accuracy can never be as good as 2C-ICE, the spatial and temporal coverage will be much larger
which is important for long time series of climate research.



*Code and data availability.* FY-3B MWHS and MWRI data can be downloaded from http://satellite.nsmc.org.cn/portalsite/. CloudSat 2C-ICE product can be downloaded from https://www.cloudsat.cira.colostate.edu/data-products. Aqua/MODIS L3 product can be downloaded from https://ladsweb.modaps.eosdis.nasa.gov/search/order. ERA5 reanalysis data can be downloaded from https://cds.climate.copernicus.eu/cdsapp#!/search?type=dataset.

*Author contribution.* Zhenzhan Wang and Wenyu Wang designed the study. Wenyu Wang performed the implementation and
wrote the manuscript. Qiurui He and Lanjie Zhang provided the training data and established the network model. Zhenzhan Wang revised the article.

*Competing interests.* The authors declare that they have no conflict of interest.

*Acknowledgements.* The authors would like to thank National Satellite Meteorological Center, China Meteorological Administration for providing the FY-3B MWHS and MWRI data. The authors thank the CloudSat and CALIPSO science teams for their hard efforts
in providing the 2C-ICE product. The authors also thank the MODIS and ERA5 teams for providing the Aqua/MODIS L3 product and ERA5 reanalysis data. We would also like to thank the reviewers and the editors for their valuable and helpful suggestions.

*Financial support.* This work was supported by the National Natural Science Foundation of China under Grant No. 41901297 and No. 42105130, the Science and Technology Key Project of Henan Province under Grant No. 202102310017.

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
