# Peer review of "Retrieval of ice water path from the FY-3B MWHS polarimetric measurements based on deep neural network"

_Atmospheric Measurement Techniques, 2022_

## Author Comment (AC1)

**Response to Reviewers**

**Reviewer 1**

We would like to sincerely thank the reviewer for his professional comments and helpful suggestions. We believe they help us to improve the manuscript significantly and give us many useful ideas to our work. We have revised the manuscript according to the reviewer's comments and answered the reviewer's question point by point below.

Reviewer comments are in italic blue and our responses in black, the manuscript changes are in red.

**Reply to General comments**

*The core idea of the manuscript, to use polarized microwave observations from the MWHS sensor to retrieve ice water path, is certainly of scientific interest given that these observations have not been used for this purpose before. In its current form, however, the manuscript lacks novelty and scientific rigor.*

*My main criticism is that the authors do very little to tie their results to any reference data, which hampers the credibility of the presented retrieval results. Although they provide a comparison of the global distributions of monthly mean IWP to ERA5, MODIS and the 2C-ICE product, the latter of which is used as training data for the retrieval, I consider these results insufficient to conclude that the retrieval works reliably given that retrieval artifacts are clearly visible over the Tibetan plateau for the winter time retrievals.*

*While I consider the topic fit for publication, major revision will be required to improve quality and relevance of the presented results.*

Reply: Thank you very much for the comments. We agree with the criticism and the manuscript has been revised and rewritten in response to reviewers' comments. The major changes in the revision are as follows:

(1) The collocation data set has been reanalyzed according to the suggestions from reviewer 1 and reviewer 2, the scatter plots were replaced by density plots; collocations in high latitudes were randomly removed to make the data set balance. (Sec. 2.2, Fig. 3-5)

(2) Most of the graphics have been revised.

(3) Network evaluation metrics were added, including FAR, POD, RMSE, MAPE, BIAS, and Pearson correlation coefficient. (Sec. 3)

(4) The networks have been re-trained using the new data set, and the classification network was redesigned using the focal loss function to reduce the effect of data imbalance. The artifacts over the Tibetan plateau have been reduced. However, the classification of ice clouds using MWHS channels is still limited, especially over the Tibetan plateau. (Sec. 4.1)

(5) Reanalysis of the new retrieval results, including the final configuration results. (Sec. 4.2)

(6) A case of tropical cyclone observed simultaneously by MWHS and CloudSat was added. (Sec. 4.3.1, Fig. 9-10)

(7) The annual mean IWP for 2015 was calculated and compared with the other three data sets. (Sec. 4.3.2, Fig. 11)

(8) Comparison of zonal means of IWP was added. (Sec. 4.3.2, Fig. 12)

**Reply to specific comments**

*Fig. 4 and 5: I would suggest analyzing only observations from the swath edge or to separate the analysis of observations from edge and center of the swath. This will make it easier to compare your results to observations from conical scanners. I also suspect the scatter plot is misleading here as many markers are likely lying on top of each other. I suggest replacing the scatter plot with a density plot. Information on the hydrometeor content can be added by drawing a contour plot of the distribution of pixels with $IWP > 10^3$ or $10^2$ on top of the density plot (with a sufficiently different colormap of course).*

Reply: Thanks for the good suggestion. The scatter plot here is indeed misleading, we have re-analysed the edge observations scan angles from ($\pm40.15°$ to $\pm53.35°$) and plotted the density. The specific changes are as follows:

(Sec. 2.2)"The density plots of the PD and TB at 150 GHz (clear-sky and cloudy scene) and the corresponding IWP from 2C-ICE over the ocean and land are depicted in Fig. 4 and 5. Scan angles from $\pm40.15°$ to $\pm53.35°$ are selected to compare the results with observations from conical scanners. In the cloudy case, the TBs are distributed between 150 K and 290 K, with the largest PD occurring at 230 K (corresponding to IWP >1000 g m$^{-2}$). This is similar to the result of Gong et al (2017, 2020). However, due to the cross-track scanning mode, the PD of MWHS is much lower than conical scanners. The lowest TB generally appears in the center of deep convection clouds, and the PD is small due to the randomly oriented ice particles; the largest PD due to the horizontally oriented particles generally appears in the warmer ice clouds. From Fig. 4, it can be seen that the lower the TB, the larger IWP, but the TB is also influenced by the local atmospheric temperature. Comparing Fig. 4 and Fig. 5, the TB of the clear sky is generally above 240 K. The PD from the ocean surface is relatively large, while the PD from land is small.

[Figure]

[Figure]

(a)                                                                 (b)

(c)                                                                 (d)

Figure 4. The PD–TB$_{150V}$ density plots for the collocations in the cloudy scenes over the ocean (a) and land (b). The (c) and (d) show the corresponding IWP from 2C-ICE.

[Figure]

(a)                                                                 (b)

Figure 5. Same as Fig. 4 but for clear-sky scenes."

*l. 177: You cannot reference a result that you haven't yet presented.*

Reply: Thank you for the comment. The sentence has been deleted.

*Fig. 5: Please elaborate what causes the low TBs for clear sky observations.*

Reply: Thanks for the comment. The low TBs in clear sky observations is mainly caused by low atmospheric temperature, occurring at high altitudes such as the Tibetan Plateau or in regions with atmospheric temperature anomalies. There is also a possibility that ice clouds exist outside the FOV of CloudSat (only one-tenth of the MWHS pixel) but in the FOV of MWHS.

*Sec. 4.2: The presentation of the evaluation needs to be improved. You need to be clearer about what data is used to calculate the error for the cases you present. In particular, you need to state whether for Cases 1 - 5*

*the changes to the input data are also applied to the validation data. If that is the case, these error metrics have little meaning as the distribution they are calculated over cannot be expected to represent that of real measurements.*

Reply: Thanks. We agree with the comment. Cases 1-5 have been deleted. The presentation is revised as follows:

"For the global IWP retrieval, clear-sky scenes were excluded from the training data. Different combinations of the network input are compared to find the best retrieval strategy. The auxiliary information cases and their retrieval errors are listed in Table 3. In these cases, five channels are all used. Additional information including latitude, scan angle and ocean/land mask and their combinations were added to train the networks. " (Sec. 4.2)

*l. 236: Define relative error. I assume you are referring to the absolute percentage error here. Note that when the absolute percentage error is used to select between models it is biased towards models that underestimate the true value. This together with the fact that you are using MSE of log(IWP) to train your network, this will likely lead to non-negligible biases in your retrieval. You should therefore also add bias to Tab. 2. I would also suggest adding correlation as an additional metric.*

Reply: Thanks for the good suggestion. The definition of errors is added to the end of Sec. 3. The network does tend to underestimate the true value. We have added RMSE, BIAS and correlation to the Tables. And we used MAE as the loss function in this version since its result is better than MSE. The changes are as follows:

(Sec. 3)"The performance metrics employed for the retrieval are defined in the following.

The commonly used binary classification metrics are chosen for the cloud filtering network. A confusion matrix M is defined as

$$M = \begin{pmatrix} TP & FP \\ FN & TN \end{pmatrix} \tag{1}$$

$TP$ and $TN$ are the number of true positives (both MWHS and CloudSat find ice clouds) and negatives (both MWHS and CloudSat find no ice clouds), respectively. $FP$ and $FN$ are the number of false positives (MWHS finds ice clouds but CloudSat not) and negatives (CloudSat finds ice clouds but MWHS not), respectively

From the confusion matrix above, the accuracy (AC), False Alarm Ratio (FAR), Probability of Detection (POD), F1 score and Critical Success Index (CSI) can be derived as

$$AC = \frac{TP + TN}{TP + TN + FP + FN} \tag{2}$$

$$FAR = \frac{FP}{TP + FP} \tag{3}$$

$$POD = \frac{TP}{TP + FN} \tag{4}$$

$$F1 = \frac{2 \cdot TP}{2 \cdot TP + FP + FN} \tag{5}$$

$$CSI = \frac{TP}{TP + FN + FP} \tag{6}$$

The performance evaluation for the IWP retrieval network is based on the RMSE, MAPE, BIAS and Pearson correlation coefficient (CC), defined as

$$RMSE = \sqrt{\frac{1}{N} \sum_{i=1}^{N} (y_{pred,i} - y_{valid,i})^2} \tag{7}$$

$$MAPE = \frac{1}{N} \sum_{i=1}^{N} \frac{|y_{pred,i} - y_{valid,i}|}{y_{valid,i}} \times 100\% \tag{8}$$

$$BIAS = \frac{1}{N} \sum_{i=1}^{N} (y_{pred,i} - y_{valid,i}) \tag{9}$$

$$CC = \frac{\frac{1}{N} \sum_{i=1}^{N} (y_{pred,i} - \overline{y_{pred}})(y_{valid,i} - \overline{y_{valid}})}{\sigma_{pred} \sigma_{valid}} \tag{10}$$

(Sec. 4.2) For the global IWP retrieval, clear-sky scenes were excluded from the training data. Different combinations of the network input are compared to find the best retrieval strategy. The auxiliary information cases and their retrieval errors are listed in Table 3. In these cases, five channels are all used. Additional information including latitude, scan angle and ocean/land mask and their combinations were added to train the networks.

Concerning the errors shown in Table 3, a significant improvement in retrieval performance is achieved by adding latitude or ocean/land mask information while the contribution of just adding the scan angle to the retrieval is not significant. In MWHS measurements, the signal from ice clouds is a reduction in TB by scattering effects. In the absence of latitude information, it is difficult to distinguish whether the decrease in TB is due to the ice particles or the low radiance from the surface or atmosphere. So is the ocean/land mask information. According to cases 1, 2, 4 in Table 3, the CC is improved from 0.50 to about 0.62, RMSE and MAPE are also improved significantly. However, MAPE and BIAS are in conflict, reducing MAPE will increase BIAS. Thus, the correlation is an important metric for evaluating the model. The combination of

auxiliaries can further improve the retrieval results, although the effect of using the scan angle alone is not obvious. Case 5 and 6 in Table 3 indicates that the scan angle combined with latitude and ocean/land mask can also further improve the retrieval capability.   The retrieval MAPE of each IWP bin is shown in Fig. 7 (a). The MAPE in different IWP bins gives a more detailed comparison. Compared to no auxiliary model, adding auxiliaries can significantly reduce the retrieval errors, especially at IWP $<200$ g m$^{-2}$ and IWP $>1000$ g m$^{-2}$.

Table 3. Errors of IWP retrieval using different auxiliaries

|  | RMSE (g m$^{-2}$) | MAPE (%) | BIAS (g m$^{-2}$) | CC |
|---|---|---|---|---|
| 1. No | 1085.75 | 109.94 | -91.09 | 0.50 |
| 2. Lat | 943.68 | 84.53 | -125.98 | 0.61 |
| 3. Ang | 1020.52 | 106.43 | -93.64 | 0.53 |
| 4. Mask | 943.80 | 81.84 | -126.03 | 0.62 |
| 5. Lat+Ang | 908.59 | 79.88 | -145.70 | 0.64 |
| 6. Lat+Mask | 908.48 | 75.80 | -141.02 | 0.64 |
| 7. Ang+Mask | 895.98 | 78.60 | -143.64 | 0.65 |
| 8. Lat+Ang+Mask | 875.20 | 75.30 | -117.05 | 0.67 |

The performance of the different channel combinations (all the auxiliary information is added) is presented in Table 4. Since the 183 GHz channels (CH. 3-5) of MHS have proved to have good sensitivity to CloudSat IWP, the influence of the 150 GHz channel and its PD is mainly focused here. The results of case 2 and 3 in Table 4 show that adding the 150 GHz window channel (CH. 2) give an improvement to all the metrics. Considering the contribution of PD in the retrieval, the results show that the addition of PD alone (case 4) contributes to the retrieval of IWP, while the combination including both H and V polarization channels has the best performance (case 1).   Figure 7 (b) illustrate the MAPE of different channels. Comparing case 3 with case 4 in Table 4, the addition of PD gives an obvious improvement in the retrieval results at IWP $>2000$ g m$^{-2}$. This conclusion is close to the analysis in Figure 4. In general, all channels of MWHS contribute to ice cloud retrieval."

Table 4. Errors of IWP retrieval using different channels

|  | RMSE (g m$^{-2}$) | MAPE (%) | BIAS (g m$^{-2}$) | CC |
|---|---|---|---|---|
| 1. CH.1-5 | 875.20 | 75.30 | -117.05 | 0.67 |
| 2. CH.2-5 | 901.84 | 76.75 | -139.49 | 0.64 |
| 3. CH.3-5 | 932.29 | 79.34 | -158.89 | 0.61 |
| 4. CH.3-5+PD | 894.08 | 79.82 | -134.88 | 0.65 |

*l. 236: It does not make sense to include the errors over land and ocean here as this is nothing that you can tune. Please move this analysis to the end of this subsection and perform it for the final retrieval configuration.*
Reply: Thanks for the comment. It has been moved to the end of this subsection.

*l. 263: This is only true if your training data set is too small to include cases with PD from the surface. Otherwise the network can easily learn to handle ambiguous inputs given that it is trained properly.*
Reply: Thanks for the comment. The sentence has been deleted.

*l. 268: While the scatter plot is useful here it is insufficient to fully characterize the retrieval. For this you should apply your full retrieval, i.e. the combination cloud classification and IWP calculation, to all pixels from January 2015. Please provide a table containing at least bias, MSE, correlation and potentially the relative error calculated for IWP > 100 g/m^2. Here you can then also assess performance over land and ocean.*
Reply: Thanks for the good suggestion. We have added the table and analysed the full retrieval results. The performance over land and ocean is also discussed. The changes are as follows:

(Sec. 4.2)"The final retrieval models (case 1 in Table 2 and case 8 in Table 3) were selected according to the metrics. Combining the cloud filtering network and the IWP retrieval network with the test data, the final results are shown in Table 5. The performance over the ocean and land is also listed. After adding the cloud filtering network, the accuracy of IWP retrieval decreased, significantly for MAPE and BIAS, and slightly for CC and RMSE. The results are better over the ocean than over land, especially the correlation. Figure 8 shows the scatter plot between MWHS IWP and 2C-ICE IWP in January 2015. The result shows relative agreement, but MWHS IWP has significant dispersion at low IWP, which may be due to the lack of sensitivity of MWHS to thin ice clouds. The final model underestimates the true value overall but overestimates it when the IWP <300 g m$^{-2}$.

Table 5. Errors of the final selected models

|  | RMSE (g m$^{-2}$) | MAPE (%) | BIAS (g m$^{-2}$) | CC |
|---|---|---|---|---|
| Final model | 916.76 | 92.90 | -213.12 | 0.65 |
| Land | 942.81 | 92.56 | -260.47 | 0.55 |
| Ocean | 908.20 | 92.76 | -196.79 | 0.69 |

[Figure]

Figure 8. Comparison between 2C-ICE and MWHS IWP. The red line represents the diagonal 1:1 line. Clear-sky scenes are not shown."

*l. 270: This error propagation doesn't make sense. Even if your relative errors would follow a Gaussian distribution your mean relative error wouldn't be an estimator of its standard deviation. This is even less the case for the median absolute error.*

Reply: Thank you for the comment. It has been deleted.

*Sec. 4.3.1: Although retrievals of cyclones are certainly scientifically interesting, the retrieval results that you provide are not very meaningful as they can't be tied to any reference value. I suggest you try to find a co-located overpass of both CloudSat and MWHS. There's a large number of CloudSat Cyclone overpasses available from https://adelaide.cira.colostate.edu/tc/tcs-50km.txt, it should be possible to find one that coincides with an overpass from MWHS within 30 minutes or so. This would allow you to compare your retrieval results to both 2C-ICE as well as the MODIS retrievals andthus add more credibility to the results presented in Sec. 4.3.2.*

Reply: Thanks for the good suggestion. A tropical cyclone Bansi is found on January 12, 2015 with the collocation time of about 3 minutes. The changes are as follows:

(Sec. 4.3.1)"A tropical cyclone Bansi observed by MWHS and CloudSat simultaneously (the time difference is about 3 minutes) on 12 January 2015 is selected for the validation of the final networks. MWHS observed TBs of the cyclone are manifested in Fig. 9. Quite low TB (as low as 150 K) can be found at 150 GHz and 183-7 GHz channels in the regions of the eyewall (the eye is not seen) and spiral rain bands which are mainly caused by the scattering of ice particles in the clouds. The 183-1 GHz and 183-3 GHz channels are strongly influenced by water vapor, the shape of the cyclone is not observable, but clear low TBs can still be seen in the eyewall and rainband. The PDs at 150 GHz ($TB_V - TB_H$), their distribution characteristics are the same

as the low TBs. The PD reaches its maximum in the anvil precipitation regions (around 5 K, consistent with the result in Fig. 4) and decrease in the remote clear-sky or cirrus regions.

[Figure]

Figure 9. Tropical cyclone Bansi on 12 January 2015 as observed with FY-3B/MWHS channels.

Applying the two neural networks trained above to the tropical cyclone, the retrieval IWPs are shown in Fig. 10 in comparison with 2C-ICE, and the retrieval errors are listed in Table 6. Due to the narrow field of view of CloudSat, a total of 21 pixels of MWHS are collocated in the tropical cyclone region. The results show that MWHS IWP has a high correlation with 2C-ICE, the MAPE and BIAS are better than that in Table 5, although the RMSE is larger, it is reasonable in tropical cyclones.

Table 6. Errors of the tropical cyclone retrieval

|  | RMSE (g m$^{-2}$) | MAPE (%) | BIAS (g m$^{-2}$) | CC |
|---|---|---|---|---|
| Bansi | 1191.3 | 77.69 | 82.07 | 0.73 |

[Figure]

Figure 10. IWP comparison of MWHS and 2C-ICE at the tropical cyclone Bansi.''

*Sec. 4.3.2: Please add a figure with the distribution of the zonal mean IWP similar to Fig. 3 in Duncan and Eriksson, 2018. This will allow for a more quantitative evaluation of the retrieval results. I also think a logarithmic color scale (as in Duncan and Eriksson 2018) would be more suitable to display global distributions of IWP in Fig. 15 and Fig 16.*

Reply:  Thanks for the good suggestion. The changes are as follows:

(Sec. 4.3.2)"Figure 11 shows the global mean IWP for 2015 from Aqua/MODIS L3 product (MYD08_M3, C61, Platnick et al., 2017), CloudSat 2C-ICE, FY-3B/MWHS retrieval and ERA5 reanalysis data set. ERA5 IWP data shown here is combined of its total column snow water (CSW) and cloud ice water (CIW) data since it differentiates between precipitating and non-precipitating ice. The overall distribution of the annual mean IWP for the four data sets is similar. The MODIS product has a significantly higher IWP than the other three products, while the ERA5 has a lower IWP overall. IWP from 2C-ICE is the same as MODIS near the equator and between ERA5 and MODIS elsewhere. Since 2C-ICE is used to train the networks, MWHS IWP is certainly approaching the 2C-ICE. The zonal means of IWP for 2015 are given in Fig. 12. The overall shape of the IWP zonal averages is fairly consistent across data sets. However, there are large differences in the overall magnitude of the IWP. These differences are particularly pronounced at mid-latitudes, especially between the MODIS product and the other three products. Compared to the IWP maps in Duncan and Eriksson (2018), this version of MODIS IWP is more similar to 2C-ICE near the equator (10°S - 10°N), but with increasing latitude, the IWP is much larger than the other products. The MWHS IWP is very close to that of 2C-ICE but lower than 2C-ICE in the mid-latitudes of the southern hemisphere. This may be due to the lack of training data in the middle and high latitudes of the southern hemisphere.

[Figure]

Figure 11. Global mean IWP maps for 2015 from MODIS, 2C-ICE, MWHS and ERA5. 2C-ICE is gridded on a 5° grid, while the other products are gridded on a 1° grid.

[Figure]

Figure 12. Zonal means of IWP for 2015."

**Technical corrections**

*l. 4: Missing space after 'Information'*

Reply: Thanks for the comment. It has been corrected.

*l. 29 - 31: You cannot conclude that individual measurements can only sense certain properties of clouds only based on their sensitivity to microphysics.*

Reply: Thanks for the comment. The sentence has been deleted.

*l. 59: ICI will only have channels up to 668 GHz*

Reply: Thanks for the comment. It has been corrected.

*l. 62 - 64: How has MWHS been 'proven to give information about IWP' if it was 'hardly analyzed information past studies'?*

Reply: Thanks for the comment. The reference (He and Zhang, 2016) use the 89 and 150 GHz channels of MWHS-I and II onboard FY-3B/C to analyze the distribution of cirrus over the Tibetan plateau. However, the study did not conduct a more detailed analysis.

*l. 68: The name is Cloud ProfilING Radar*

Reply: Thanks for the comment. It has been corrected.

*Fig. 2 (a): Are there really gaps in the distribution or is that an artifact of the bin boundaries? If the former please explain what could cause them. In the case of the latter please select the bin boundaries to avoid them.*

Reply: Thanks for the comment. This is an artifact of the bin boundaries and the figure has been corrected.

*l. 167: Please typeset unit according to manuscript preparation guidelines.*

Reply: Thanks for the comment. The manuscript has been revised according to the guidelines.

*l. 170: Figure should be abbreviated with Fig. except at the beginning of a sentence.*

Reply: Thanks for the comment. It has been corrected.

*l. 222: Please also provide false alarm rate and probability of detection since accuracy alone can be misleading for imbalanced datasets.*

Reply: Thanks for the good suggestion. The FAR and POD have been provided in Sec. 4.1. The changes are as follows:

(Sec. 4.1)"The results show that all channels have cloud information, and CH. 4 (183-3 GHz) is the best for cloud detection. This channel is also used by the traditional method to distinguish cloudy from clear sky. However, the detection of ice clouds using MWHS channels is still limited. The FAR and POD of the best network are 0.26 and 0.63, respectively."

Table 2. Errors of cloud filtering using different channels

|  | AC | FAR | POD | F1 | CSI |
|---|---|---|---|---|---|
| 1. CH.1-5 | 0.92 | 0.26 | 0.63 | 0.67 | 0.51 |
| 2. CH.2-5 | 0.92 | 0.27 | 0.61 | 0.66 | 0.49 |
| 3. CH.3-5 | 0.91 | 0.30 | 0.62 | 0.65 | 0.49 |
| 4. CH.4-5 | 0.91 | 0.29 | 0.59 | 0.64 | 0.48 |
| 5. CH.5 | 0.83 | 0.32 | 0.38 | 0.49 | 0.33 |

*l. 289: Specify channel in which the low TB are observed*

Reply: Thanks for the comment. The changes are as follows:

(Sec. 4.3.1)"Quite low TB (as low as 150 K) can be found at 150 GHz and 183-7 GHz channels in the regions of the eyewall (the eye is not seen) and spiral rain bands which are mainly caused by the scattering of ice particles in the clouds. The 183-1 GHz and 183-3 GHz channels are strongly influenced by water vapor, the shape of the cyclone is not observable, but clear low TBs can still be seen in the eyewall and rainband."

*l. 325: showed -> shown*

Reply: Thanks for the comment. It has been corrected.

*l. 341: The different measurement resolution of Modis and MWHS cannot affect the retrieved mean on a 5x5 degree grid.*

Reply: Thanks for the comment. The previous version of the manuscript had an error in the geographical location of the MODIS map, which has been fixed.

*l. 344: Given that there are obvious artifacts in the retrieval results, I don't think that this can be concluded.*

Reply: Thanks for the comment. The sentence has been deleted.

*l. 365: This discussion of the limitations of the neural network retrieval is too superficial. First of all, once the code is written extracting more co-locations is extremely easy, so I don't think there is a valid excuse to use a training data set that oneself deems too small. Moreover, although there are uncertainties related to the co-locations of the CloudSat and MWHS observations, these uncertainties are represented in the training data and can thus be predicted using for example quantile regression neural networks. The real issue are the uncertainties in the 2C-ICE data as these are much harder to quantify and cannot be predicted.*

Reply: Thanks for the good suggestion. The changes are as follows:

(Sec. 4.4)"However, there are some limitations of using neural networks for IWP retrieval. Collocation is the first limitation since there are some uncertainties in the field of view of MWHS and CloudSat due to the large resolution difference. These uncertainties are represented in the training data and can be predicted using

for example quantile regression neural networks. The most important issue is the real sample (2C-ICE) used in training, which has uncertainties that are difficult to quantify. Therefore, it is also impossible to make accurate error estimates of the model results. In the absence of access to a large number of real samples, the use of neural networks can only converge to a certain product with the highest accuracy (such as 2C-ICE). An alternative approach is to use simulation results (typical profiles) of radiative transfer models, where the generalization ability of the network will strongly depend on the model itself and the input field. In addition, the microwave band below 200 GHz is sensitive only to large ice particles and thick clouds and is relatively less effective for cloud detection."

*l. 385: You cannot conclude that performance is good for the Cyclone cases because you don't have any reference to compare to.*

Reply: Thanks for the comment. The comparison has been added.

---

## Author Comment (AC2)

**Response to Reviewers**

**Reviewer 2**

We would like to sincerely thank the reviewer for his professional comments and helpful suggestions. We believe they help us to improve the manuscript significantly and give us many useful ideas to our work. We have revised the manuscript according to the reviewer's comments and answered the reviewer's question point by point below.

Reviewer comments are in italic blue and our responses in black, the manuscript changes are in red.

**Reply to general comments**

*The paper develops the Neural Network for the FY/MWHS to retrieve global IWP parameters using the correlations between the CloudSat 2C-ICE IWP products and the MWHS BT measurements when collocations happen. Although the statistical NN method does not use any forward model calculations to involve physical radiative transfer processes, it provides a simple and quick way to obtain the global IWP coverage from the FY/MWHS observations. However, the entire methods including finding collocations, developing NN, and analyzing results have so many similarities to the paper in Holl et al., 2010; 2014, which raises concerns about the significance and novelty of this work. Further, major issues as summarized below exist in the developed methods. Due to these weaknesses, the manuscript is recommended to be rejected in the present form.*

Reply: Thanks for the comments. The idea of the manuscript is to use polarized microwave observations from the MWHS to retrieve ice water path because these observations have not been used for this purpose before. We agree the criticism and the manuscript has been revised and rewritten in response to reviewers' comments. The major changes in the revision are as follows:

(1) The collocation data set has been reanalyzed according to the suggestions from reviewer 1 and reviewer 2, the scatter plots were replaced by density plots; collocations in high latitudes were randomly removed to make the data set balance. (Sec. 2.2, Fig. 3-5)

(2) Most of the graphics have been revised.

(3) Network evaluation metrics were added, including FAR, POD, RMSE, MAPE, BIAS, and Pearson correlation coefficient. (Sec. 3)

(4) The networks have been re-trained using the new data set, and the classification network was redesigned using the focal loss function to reduce the effect of data imbalance. The artifacts over the Tibetan plateau have been reduced. (Sec. 4.1)

(5) Reanalysis of the new retrieval results, including the final configuration results. (Sec. 4.2)

(6) A case of tropical cyclone observed simultaneously by MWHS and CloudSat was added. (Sec. 4.3.1, Fig. 9-10)

(7) The annual mean IWP for 2015 was calculated and compared with the other three data sets. (Sec. 4.3.2, Fig. 11)

(8) Comparison of zonal means of IWP was added. (Sec. 4.3.2, Fig. 12)

**Reply to specific comments**

*Section 2.2, lines 133-144*

*The procedure of finding collations is one of the key steps in the entire study, but the descriptions shown in lines 133-144 in Sect 2.2 are very vague. For example, the MWHS footprint size should change with the scanning angles since the instrument has a cross-track scanning mode. How accurate is it to always approximate the MWHS footprint using a constant circular pixel? How do you address the spatial response function inside each MWHS footprint? Also, when the MWHS scanning angle is too large, its field of view is likely to be different to that of the nadir-looking CPR even though the two sensors have similar geolocations, and readers might wonder how reprehensive the collations are in such situations. The sampling errors due to insufficient CPR pixels in each MWHS measurement are mentioned, but it is still not clear how significant the errors are and what the authors did to minimize the negative effects.*

Reply: Thanks for the comment.

(1) Using a constant circular pixel to approximate the MWHS footprint has no impact on collocation. The collocations at edge angles are few, and although the MWHS pixel becomes larger with scanning, the CPR is limited. The geographic distance of 7.5 km is used as a threshold to obtain as many as possible collocations with sample representativeness. If the collocation window continues to increase or decrease, the distribution of collocations will be limited to a specific spatial and time range.

(2) The spatial response function is not known, and thus we average the CPR IWP in the MWHS pixel to improve representativeness.

(3) The field of view of MWHS and CPR is different, and when the scan angle is large, the atmospheric path will be much longer than the nadir observation. However, the effect is not significant for ice clouds that typically exist in the upper troposphere. We have added the scan angle in the auxiliary parameters of the neural network, which is similar to common practice in data assimilation.

(4) There is no accurate method to evaluate the sampling error because we do not know what is happening in the MWHS pixel other than CPR pixels, and there is no other reference. This is a limitation of the collocation method. To reduce this effect, we chose collocations with the number of CPR pixels larger than 10 and the coefficient of variation less than 0.6, i.e., when a scene contains more than 10 CPR pixels within a scene and the IWP variation is relatively small, the scene is considered representative.

The changes are as follows:

"Since the collocation error cannot be estimated, the criteria discussed in Holl et al. (2010) is applied to reduce the sampling effect of collocations. In this study, an MWHS pixel with more than 10 pixels of 2C-ICE and less than 0.6 coefficients of variation were selected for subsequent processing. However, in the case of highly inhomogeneous clouds existing outside the CloudSat field of view, larger uncertainty for the IWP within MWHS pixels cannot be eliminated. After the reduction of inhomogeneous collocations, 665519 collocations were retained." (Sec. 2.2)

*Section 2.2, figures 1-3*

*The random IWP cases in the collocation database essentially represent our prior knowledge about the ice cloud distribution. Considering that the topic of this paper is to address the global water path distribution, the collation database is expected to sample the global IWP coverage without biases. The results in fig.3, however, show that the latitudinal distribution of the dataset is highly ununiform. This suggests improper weights are given to the random database cases, and therefore systematic biases are introduced during NN retrievals. How to make the collocation database cases to distribute according to our prior knowledge needs to be addressed.*

Reply: Thanks for the comment. The data set has been optimized for the latitudinal distribution. We also use the "focal loss" function to resolve the data situation with more sunny days and fewer cloudy days. The changes are as follows:

(Sec. 2.2)"Due to the high number of collocations near the poles, 121500 observations at high latitude were randomly excluded to obtain a balanced data set. For IWP retrieval, collocations should be classified into two bins (clear-sky scene and cloudy scene) according to a specific IWP threshold. A threshold of IWP >100 g m$^{-2}$ is preliminarily selected to classify cloudy scenes. Thus, 81490 collocations are recognized to be cloudy scenes and 462529 collocations are clear sky scenes in this data set."

[Figure]

(a)                                        (b)

Figure 2. Statistical information of scan angle and latitude of MWHS observations in the collocation data set.

[Figure]

Figure 3. MWHS measurements distribution of time and latitude in the collocation data set."

*Section 2.2, figures 4 and 5*
*The biggest problem of this study is the dramatic lack of validation and evaluation of the essential*
*collocation database. Since there are so many error sources in collocating, adequate work on validating the*
*dataset and evaluating the mismatch errors is necessary. The only results serving such purposes are given in*
*figures 4 and 5, but the results are very confusing. The figures show that the BT observations spread over*
*identical ranges no matter the ice clouds exist or not, at least in the way the authors show. How could*
*you retrieve ice cloud parameters if the BT observations do not respond to the ice cloud change at all?*
*Besides, I suspect that the collocation database should have many physically unreasonable cases due to*
*various error sources, right? If so, the method to filter out the meaningless cases needs to be illustrated.*
*Also, the effects of various error sources on the database and the retrieval accuracies need to be thoroughly*
*analyzed. Overall, solid evidence must be provided to assure the critical collocation database is robust.*
Reply: Thanks for the comment. We agree with your criticism.

(1)The scatter plot here is misleading, we have re-analysed the edge observations scan angles from (±40.15°

to ±53.35°) and plotted the density as suggested by reviewer 1 (Fig. 4 and 5). It can be seen from the plot that

there is a clear TB response to the ice clouds. Another problem is the appearance of low TB in clear sky in

some cases. It is mainly caused by low atmospheric temperature, occurring at high altitudes such as the Tibetan

Plateau or in regions with atmospheric temperature anomalies. Thus, latitude is important auxiliary

information in training.

(2)For physically unreasonable, there is a possibility that ice clouds exist outside the FOV of CloudSat

(only one-tenth of the MWHS pixel) but are still in the FOV of MWHS, which can also lead to low TB in

clear sky conditions. Unfortunately, we don't have access to this information.

(3)Current studies based on multiple sensors usually use the similar collocation method, such as the

CloudSat-GPM coincidence dataset 2B-CSATGPM that is widely used in cloud and precipitation studies

(Turk, 2017; Turk et al., 2021). The accuracy of collocation is still a challenging problem, especially for the joint use of active radar and passive radiometers.

This issue is also mentioned in Gong et al., 2014 and Gong et al., 2020:

"Ice cloud misclassification is an unavoidable issue for any cloud retrieval technique. Cloud misclassification of this retrieval algorithm is partly induced by the beam-filling effect and mismatch of CloudSat and MHS footprints spatially and temporarily." (Gong et al., 2014)

"On the other hand, since match-up is defined to happen whenever the CPR beam intercepts with the DPR beam at any altitude and at any DPR view angle, the line-of-sight volume is quite different when DPR is at an off-nadir view angle, and this problem is even more severe for GMI which always views at a slant angle. Even though a cosine function is multiplied to slightly mitigate this issue (Turk, 2017), 3D cloud inhomogeneity and beam-filling effects are again the culprit of uncertainty that is hard to justify." "In our case, footprint size and line-of-sight mismatch are likely the largest sources of bias/uncertainty due to imperfect match." (Gong et al., 2020)

However, just as reviewer 1 said: "Although there are uncertainties related to the co-locations of the CloudSat and MWHS observations, these uncertainties are represented in the training data." Although the processing of neural networks is a black box, it is able to identify these features.

In addition, since there are no accurate in-suit measurements of the ice clouds, our study aims to explore the hidden ice cloud information in the previously neglected microwave radiometer measurements, which has implications for the development and selection of future satellite missions. The results of the comparison between annual and latitudinal means from MWHS, MODIS, 2C-ICE and ERA5 show that the data set is reliable.

**Reference:**

Turk, F. J. CloudSat-GPM coincidence dataset, Algorithm and Theoretical Basis Document, available at: https://pps.gsfc.nasa.gov/Documents/CSATGPM_COIN_ATBD.pdf, 2017.

Turk, F. J., Ringerud, S. E., Camplani, A, et al. Applications of a CloudSat-TRMM and CloudSat-GPM Satellite Coincidence Dataset. Remote Sensing., 13(12), 2264, https://doi.org/10.3390/rs13122264, 2021.

Gong, J., Wu, D. L. CloudSat-constrained cloud ice water path and cloud top height retrievals from MHS 157 and 183.3 GHz radiances, Atmos. Meas. Tech., 7(6), 1873-1890, https://doi.org/ 10.5194/amt-7-1873-2014, 2014.

Gong, J., Zeng, X., Wu, D. L., et al. Linkage among ice crystal microphysics, mesoscale dynamics, and cloud and precipitation structures revealed by collocated microwave radiometer and multifrequency radar observations, Atmos. Chem. Phys., 20, 12633–12653, https://doi.org/10.5194/acp-20-12633-2020, 2020.

The changes are as follows:

(Sec. 2.2)"The density plots of the PD and TB at 150 GHz (clear-sky and cloudy scene) and the corresponding IWP from 2C-ICE over the ocean and land are depicted in Fig. 4 and 5. Scan angles from ±40.15° to ±53.35° are selected to compare the results with observations from conical scanners. In the cloudy case, the TBs are distributed between 150 K and 290 K, with the largest PD occurring at 230 K (corresponding to IWP >1000 g m$^{-2}$). This is similar to the result of Gong et al (2017, 2020). However, due to the cross-track scanning mode, the PD of MWHS is much lower than conical scanners. The lowest TB generally appears in the center of deep convection clouds, and the PD is small due to the randomly oriented ice particles; the largest PD due to the horizontally oriented particles generally appears in the warmer ice clouds. From Fig. 4, it can be seen that the lower the TB, the larger IWP, but the TB is also influenced by the local atmospheric temperature. Comparing Fig. 4 and Fig. 5, the TB of the clear sky is generally above 240 K. The PD from the ocean surface is relatively large, while the PD from land is small.

[Figure]

(a)                                                                           (b)

(c)                                                                           (d)

Figure 4. The PD–TB$_{150V}$ density plots for the collocations in the cloudy scenes over the ocean (a) and land (b). The (c) and (d) show the corresponding IWP from 2C-ICE.

[Figure]

Figure 5. Same as Fig. 4 but for clear-sky scenes."

Finally, the reliability of this study was verified using the annual mean IWP map and latitudinal mean plot compared with MODIS, 2C-ICE and ERA5.

(Sec. 4.3.2)"Figure 11 shows the global mean IWP for 2015 from Aqua/MODIS L3 product (MYD08_M3, C61, Platnick et al., 2017), CloudSat 2C-ICE, FY-3B/MWHS retrieval and ERA5 reanalysis data set. ERA5 IWP data shown here is combined of its total column snow water (CSW) and cloud ice water (CIW) data since it differentiates between precipitating and non-precipitating ice. The overall distribution of the annual mean IWP for the four data sets is similar. The MODIS product has a significantly higher IWP than the other three products, while the ERA5 has a lower IWP overall. IWP from 2C-ICE is the same as MODIS near the equator and between ERA5 and MODIS elsewhere. Since 2C-ICE is used to train the networks, MWHS IWP is certainly approaching the 2C-ICE. The zonal means of IWP for 2015 are given in Fig. 12. The overall shape of the IWP zonal averages is fairly consistent across data sets. However, there are large differences in the overall magnitude of the IWP. These differences are particularly pronounced at mid-latitudes, especially between the MODIS product and the other three products. Compared to the IWP maps in Duncan and Eriksson (2018), this version of MODIS IWP is more similar to 2C-ICE near the equator (10°S - 10°N), but with increasing latitude, the IWP is much larger than the other products. The MWHS IWP is very close to that of 2C-ICE but lower than 2C-ICE in the mid-latitudes of the southern hemisphere. This may be due to the lack of training data in the middle and high latitudes of the southern hemisphere.

[Figure]

Figure 11. Global mean IWP maps for 2015 from MODIS, 2C-ICE, MWHS and ERA5. 2C-ICE is gridded on a 5° grid, while the other products are gridded on a 1° grid.

[Figure]

Figure 12. Zonal means of IWP for 2015."

*Section 4.2*

*Figures. 7 to 10 and table 2 show the statistics of the retrieval results using different inputs, and they are the primary results of this paper. However, the results become unpersuasive since the collocation database is not established, validated, and evaluated properly. Lines 270-273 give the estimations of the retrieval errors by*

*combining the 2C-ICE product errors and the NN retrieval errors, but the errors from the collocation finding procedure are not considered. The testing dataset is obtained in the same way as the training dataset, which means the two datasets share the same inherent collocation errors. Besides, no descriptions and explanations of fig.10 are given, and more discussions should be added in the revision.*

Reply: Thanks for the comment. The issue of collocations has been discussed above. Additional evaluation metrics have been added here based on the suggestion from reviewer 1. We have added the table and analysed the full retrieval results. The performance over land and ocean is also discussed. The changes are as follows:

(Sec. 4.2)"The results show that all channels have cloud information, and CH. 4 (183-3 GHz) is the best for cloud detection. This channel is also used by the traditional method to distinguish cloudy from clear sky. However, the detection of ice clouds using MWHS channels is still limited. The FAR and POD of the best network are 0.26 and 0.63, respectively." (Sec. 4.1)

Table 2. Errors of cloud filtering using different channels

|  | AC | FAR | POD | F1 | CSI |
|---|---|---|---|---|---|
| 1. CH.1-5 | 0.92 | 0.26 | 0.63 | 0.67 | 0.51 |
| 2. CH.2-5 | 0.92 | 0.27 | 0.61 | 0.66 | 0.49 |
| 3. CH.3-5 | 0.91 | 0.30 | 0.62 | 0.65 | 0.49 |
| 4. CH.4-5 | 0.91 | 0.29 | 0.59 | 0.64 | 0.48 |
| 5. CH.5 | 0.83 | 0.32 | 0.38 | 0.49 | 0.33 |

Table 3. Errors of IWP retrieval using different auxiliaries

|  | RMSE (g m$^{-2}$) | MAPE (%) | BIAS (g m$^{-2}$) | CC |
|---|---|---|---|---|
| 1. No | 1085.75 | 109.94 | -91.09 | 0.50 |
| 2. Lat | 943.68 | 84.53 | -125.98 | 0.61 |
| 3. Ang | 1020.52 | 106.43 | -93.64 | 0.53 |
| 4. Mask | 943.80 | 81.84 | -126.03 | 0.62 |
| 5. Lat+Ang | 908.59 | 79.88 | -145.70 | 0.64 |
| 6. Lat+Mask | 908.48 | 75.80 | -141.02 | 0.64 |
| 7. Ang+Mask | 895.98 | 78.60 | -143.64 | 0.65 |
| 8. Lat+Ang+Mask | 875.20 | 75.30 | -117.05 | 0.67 |

Table 4. Errors of IWP retrieval using different channels

|  | RMSE (g m⁻²) | MAPE (%) | BIAS (g m⁻²) | CC |
|---|---|---|---|---|
| 1. CH.1-5 | 875.20 | 75.30 | -117.05 | 0.67 |
| 2. CH.2-5 | 901.84 | 76.75 | -139.49 | 0.64 |
| 3. CH.3-5 | 932.29 | 79.34 | -158.89 | 0.61 |
| 4. CH.3-5+PD | 894.08 | 79.82 | -134.88 | 0.65 |

The final retrieval models (case 1 in Table 2 and case 8 in Table 3) were selected according to the metrics. Combining the cloud filtering network and the IWP retrieval network with the test data, the final results are shown in Table 5. The performance over the ocean and land is also listed. After adding the cloud filtering network, the accuracy of IWP retrieval decreased, significantly for MAPE and BIAS, and slightly for CC and RMSE. The results are better over the ocean than over land, especially the correlation. Figure 8 shows the scatter plot between MWHS IWP and 2C-ICE IWP in January 2015. The result shows relative agreement, but MWHS IWP has significant dispersion at low IWP, which may be due to the lack of sensitivity of MWHS to thin ice clouds. The final model underestimates the true value overall but overestimates it when the IWP <300 g m⁻².

Table 5. Errors of the final selected models

|  | RMSE (g m⁻²) | MAPE (%) | BIAS (g m⁻²) | CC |
|---|---|---|---|---|
| Final model | 916.76 | 92.90 | -213.12 | 0.65 |
| Land | 942.81 | 92.56 | -260.47 | 0.55 |
| Ocean | 908.20 | 92.76 | -196.79 | 0.69 |

[Figure]

Figure 8. Comparison between 2C-ICE and MWHS IWP. The red line represents the diagonal 1:1 line. Clear-sky scenes are not shown."

*Section 4.3.1*

*Figures 11 to 14 show a case study to retrieve IWP of the typhoon Rammasun using MWHS measurements. Again, validations of the retrieval results are completely missing. The statements say that "the structure and the distribution of IWP are consistent with the characteristic of TB (line 324)" and therefore "the performance of the two neural networks appears to be good (line 328)", which are very crude. Besides, the atmospheric and cloud microphysical statistics in typhoon are likely to be very dissimilar to the globally averaged microphysics in the collocation database. Using a different training dataset with more accurate prior information should make the typhoon retrievals better. Also, the plots of BT measurements in figures 11 -13 occupy too much space. You should provide more analytical results instead of merely showing the instrument observations.*

Reply: Thanks for the good suggestion. We believe that the results would certainly be better if a dedicated typhoon data set is used. A tropical cyclone Bansi observed by MWHS and CloudSat simultaneously is found on January 12, 2015 with the collocation time of about 3 minutes. It is used to validate the performance of the network in tropical cyclone case. The plots of BT measurements have been removed. The modifications are as follows:

(Sec. 4.3.1)"A tropical cyclone Bansi observed by MWHS and CloudSat simultaneously (the time difference is about 3 minutes) on 12 January 2015 is selected for the validation of the final networks. MWHS observed TBs of the cyclone are manifested in Fig. 9. Quite low TB (as low as 150 K) can be found at 150 GHz and 183-7 GHz channels in the regions of the eyewall (the eye is not seen) and spiral rain bands which are mainly caused by the scattering of ice particles in the clouds. The 183-1 GHz and 183-3 GHz channels are strongly influenced by water vapor, the shape of the cyclone is not observable, but clear low TBs can still be seen in the eyewall and rainband. The PDs at 150 GHz ($TB_V - TB_H$), their distribution characteristics are the same as the low TBs. The PD reaches its maximum in the anvil precipitation regions (around 5 K, consistent with the result in Fig. 4) and decrease in the remote clear-sky or cirrus regions.

[Figure]

Figure 9. Tropical cyclone Bansi on 12 January 2015 as observed with FY-3B/MWHS channels.

Applying the two neural networks trained above to the tropical cyclone, the retrieval IWPs are shown in Fig. 10 in comparison with 2C-ICE, and the retrieval errors are listed in Table 6. Due to the narrow field of view of CloudSat, a total of 21 pixels of MWHS are collocated in the tropical cyclone region. The results show that MWHS IWP has a high correlation with 2C-ICE, the MAPE and BIAS are better than that in Table 5, although the RMSE is larger, it is reasonable in tropical cyclones.

Table 6. Errors of the tropical cyclone retrieval

|  | RMSE (g m$^{-2}$) | MAPE (%) | BIAS (g m$^{-2}$) | CC |
|---|---|---|---|---|
| Bansi | 1191.3 | 77.69 | 82.07 | 0.73 |

[Figure]

Figure 10. IWP comparison of MWHS and 2C-ICE at the tropical cyclone Bansi."

*At last, there are many grammatical errors, and an English revision is necessary to improve the manuscript.*
Reply: Thanks for the comment. We have rechecked the grammar. However, due to lack of time, English revision will be performed by professional editor later.

---

## Referee Report (RR1)

**Review 2**

I would like to compliment the authors on their efforts to improve the manuscript. I think that the additional validation adds to the credibility of the proposed retrieval. Especially the good agreement with the 2C-ICE dataset is encouraging.

However, I think that the manuscript can still be improved in terms of scientific novelty and clarity of the presentation.

**General comments**

In terms of focus, I think the authors need to make their conclusions more specific. Currently, their main conclusion is that adding the 150 GHz channels to the retrieval improves its accuracy, which is not very informative. I therefore suggest to include the retrievals which use only channels around 183 GHz and retrieval using Ch. 2-5 in the results in section 4.3. This would allow the authors to quantify the effects of using one or two 150 GHz channels on the instantaneous accuracy (using the tropical cyclone case) as well as the climatological accuracy (using the comparison of the yearly means). This would put the results into a more practical context and provide the novelty required for publication in AMT.

I would also like to encourage the authors to publish their code and include a reference to it in the manuscript.

**Specific comments**

- l. 32: get microphycial of clouds -> determine the bulk and microphysical properties of clouds

- l. 48: microphysical -> microphysical properties

- l. 67: The name of the sensor is Cloud Profiling Radar. I added the capitalization just to emphasize the difference. I am sorry if this caused confusion.

- l. 74: ... based on the deep neural network -> ... based on a deep neural network

- l. 128: Ice clouds are ...

- l. 129: I would suggest using phenomenon or component instead of parameter

- l. 177: Why would the PD at similar incidence angles as that of conical scanners be lower? Isn't that rather an effect of the much larger footprint?

- Fig. 4 (c) and (d): Please reduce the bin size of for these plots or smooth the results to make the contours less noisy.

- l. 193: ... a nonlinear mapping from the input to the output data

- l. 250: If I understand you correctly, the conclusion that Ch. 4 is the best for cloud detection is based on it adding the largest improvement in Tab. 2. I don't think that this is a valid conclusion as it may just be that only Ch. 5 and Ch. 3 together work better than Ch. 4 - 5.

- l. 259: . . . five channels are all used -> . . . all five channels are used

- l. 276: . . . focused on here.

- l. 309: The PDs at . . . : Some text seems to be missing here.

- l. 309: The PDs do not share the distribution characteristics of the low TBs. That would be in contradiction of PDs occurring mostly in warmer ice clouds and outflow regions.

- l. 379: Refine conclusions to assess effects of a single 150 GHz channels as well as both 150 Ghz in terms of instantaneous estimates as well as annual means.

- Fig. 10: Please change the comparison to 2C-ICE to a line plot include this as an additional figure. I suggest you also add results from a retrieval using only channels around 183 as well as a retrieval using Ch. 2 - 5.

- Fig. 12: Here I suggest you also include results from a retrieval using only the channels around 183 as well as a retrieval using Ch. 2 - 5.

- l. 361: If you mention quantile regression neural networks, please cite Pfreundschuh et al. 2018.

- l. 376: See comment above on impact of Ch. 4

- l. 379: Refine conclusions to assess effects of a single 150 GHz channels as well as both 150 Ghz in terms of instantaneous estimates as well as annual means.

---

## Referee Report (RR2)

**Review 3**

Once again, I would like to compliment the authors on the improvements in this most recent iteration. I have only found a few smaller, mostly language-related issues. Since I am not a native speaker myself please take the following suggestions with a grain of salt.

**Comments**

- l. 11: . . . in its retrieval.

- l. 19: . . . of inputs of auxiliary variables and . . .

- l. 24: . . . in the midlatitudes . . .

- l. 91: The scanning is *across* the orbit, isn't it?

- l. 241 . . . the training data (or collocations) are well representative.

- l. 290: "So is the ocean/land mask information.". I guess you want to express that the land/ocean mask helps the retrieval distinguish hydrometeor scattering from the effect of the surface. However, that's not how I understand this sentence. Please consider reformulating it.

- l. 306: I do not see the connection to Fig. 7. Do you mean Fig. 10?

- This is really a detail, but please consider using vector graphics for all line plots.

- Fig. 7: Please add y-axis labels to all plots or at least the first plot of every row.

---

## Referee Report (RR3)

The manuscript has been revised based on reviewers' comments, and several new figures have been added. However, more devastating flaws have been exhibited through the new figures, indicating that the developed NN algorithm simply cannot do the global IWP retrievals as the authors claim.

1. The biggest issue could be found in the IWP-TB relationship in figure 4. Since the topic of this study is to perform the global IWP retrievals, the NN training database is required to cover the entire possibilities in the measurement space. I did quick forward model simulations using the mid-latitude atmosphere/cloud profiles, and the IWP-TB relationship I get is shown below. The Comparison of these two figures indicates that the collocation dataset only captures a very small fraction of the possible TB range, especially when the IWP is over 100 g/m$^2$. Similar conclusions can be drawn by comparing figure 4 and the MWHS measurement of the tropical cyclone in figure 12. For instance, the lowest TB of 183+-1GHz channel in the center of the cyclone reaches 180K, but the smallest TB value of the same channel in the collocation database is around 230K. The NN is impossible to handle such a level of extrapolations. The global TB measurements used in section 4.3.2 are not given, and I believe there must be considerable amounts of TB measurements that are out of the coverage of the collocation database.

[Figure]

The IWP-TB relationship obtained from forward model simulations using mid-latitude atmosphere/cloud profiles.

2. Figure 9 is not what I asked when I suggested investigating the sparsity of the measurement space in the last round of review. The training dataset and the validation dataset are both split from the collocation dataset, and there is no doubt they share the same statistics. What I intend to see is the comparison of TB in the training/validation database versus the cyclone

and the global TB measurements applied in section 4.3. As discussed above, figure 4 shows that the collocation database is far from fully covering the TB space, and therefore we cannot expect the NN to produce sensible retrieval results.

3. Another critical issue is the retrieval experiment in figure 11, which tests the NN retrieval accuracies by comparing the retrieved parameters with the reference IWP using a testing database. The testing dataset is obtained from the same collocation finding procedure as the training/validation dataset but over a different time. Although the testing dataset is not used in the training, a well-established NN is capable to produce very accurate results since the testing and training/validation datasets have very similar statistics. However, figure 11 shows the correlations between the retrieved IWP and the reference are terrible when IWP is smaller than 1 kg/m$^2$. The authors say this "may be due to the lack of sensitivity of the MWHS to thin ice clouds" (line 319), but even figure 4 shows that the MWHS channels are sensitive to the IWP when it is over 100 g/m$^2$. Figure 8 in Holl et al., 2014 conducted an identical testing experiment, and the NN results they got are consistent with the 2C-ICE along the whole range, which is in line with expectations. The training/validation dataset is undoubtedly one contributing factor to the poor performance, but we cannot eliminate the possibility that the NN is not appropriately implemented.

In summary, I do not believe the developed NN algorithm has the capability to perform the global IWP retrievals and support the quantitative conclusions the authors claim. The idea of using collocations to train NN is great, but fundamental improvements and validations are required before this algorithm can be applied in practice.

---

## Author Response (AR2)

**Response to Reviewers**

**Reviewer 1**

We would like to sincerely thank the reviewers for their professional comments and helpful suggestions. We believe they help us to improve the manuscript significantly and provide many useful ideas to our work. This manuscript has been revised by native English editors. We have revised the manuscript according to the reviewer's comments and answered the reviewer's question point by point below.

Reviewer comments are in italic blue and our responses in black, the manuscript changes are in red.

**Reply to General comments**

*In terms of focus, I think the authors need to make their conclusions more specific. Currently, their main conclusion is that adding the 150 GHz channels to the retrieval improves its accuracy, which is not very informative. I therefore suggest to include the retrievals which use only channels around 183 GHz and retrieval using Ch. 2-5 in the results in section 4.3. This would allow the authors to quantify the effects of using one or two 150 GHz channels on the instantaneous accuracy (using the tropical cyclone case) as well as the climatological accuracy (using the comparison of the yearly means). This would put the results into a more practical context and provide the novelty required for publication in AMT. I would also like to encourage the authors to publish their code and include a reference to it in the manuscript.*

Reply: Thanks for the great suggestions. The manuscript has been revised and rewritten in response to reviewers' comments. The comparisons of adding 150 GHz channel and only using 183 GHz channels in tropical cyclone and year maps have been added. In addition, the code and test data has been uploaded to Zenodo and mentioned in the manuscript.

**Reply to specific comments**

*• l. 32: get microphycial of clouds -> determine the bulk and microphysical properties of clouds*

Reply: Thanks for the comment. It has been corrected.

*• l. 48: microphysical -> microphysical properties*

Reply: Thanks for the comment. It has been corrected.

*• l. 67: The name of the sensor is Cloud Profiling Radar. I added the capitalization just to emphasize the difference. I am sorry if this caused confusion.*

Reply: Sorry, I misunderstood your comment before. It has been corrected now. Thanks again.

*• l. 74: . . . based on the deep neural network -> . . . based on a deep neural network*

Reply: Thanks for the comment. It has been corrected.

*• l. 128: Ice clouds are . . .*

Reply: Thanks for the comment. It has been corrected.

*• l. 129: I would suggest using phenomenon or component instead of parameter*

Reply: Thanks for the comment. It has been corrected.

*• l. 177: Why would the PD at similar incidence angles as that of conical scanners be lower? Isn't that rather an effect of the much larger footprint?*

Reply: Thanks for the comment. I mean the PD of the quasi-polarization channels is lower than the real PD. The much larger footprint of MWHS is of course an important reason. The changes are as:

"However, due to the quasi-polarization mode and the much larger footprint, the PD of MWHS is much lower than that of conical scanners (e.g. GMI)."

*• Fig. 4 (c) and (d): Please reduce the bin size of for these plots or smooth the results to make the contours less noisy.*

Reply: Thanks for the comment. The figure has been changed as:

[Figure]

*• l. 193: . . . a nonlinear mapping from the input to the output data*

Reply: Thanks for the comment. It has been corrected.

*• l. 250: If I understand you correctly, the conclusion that Ch. 4 is the best for cloud detection is based on it adding the largest improvement in Tab. 2. I don't think that this is a valid conclusion as it may just be that only Ch. 5 and Ch. 3 together work better than Ch. 4 - 5.*

Reply: Thanks for the comment. The conclusion is corrected and the Table 2 has been revised. The changes are as:

"The cloud filtering performance for different channel combinations is listed in Table 2. The results showed that all three 183 GHz channels have cloud identification capability, and the addition of one 150 GHz channel enhances the POD of the network, while the two 150 GHz channels do not yield additional information."

Table 2. Errors of cloud filtering using different channels

| | AC | FAR | POD | F1 | CSI |
|---|---|---|---|---|---|
| 1. CH.1-5 | 0.91 | 0.31 | 0.61 | 0.65 | 0.48 |
| 2. CH.2-5 | 0.91 | 0.31 | 0.61 | 0.65 | 0.48 |
| 3. CH.3-5 | 0.91 | 0.31 | 0.54 | 0.60 | 0.43 |

| | | | | | |
|---|---|---|---|---|---|
| 4. CH.3&4 | 0.90 | 0.30 | 0.52 | 0.59 | 0.42 |
| 5. CH.3&5 | 0.90 | 0.31 | 0.50 | 0.58 | 0.41 |
| 6. CH.4&5 | 0.91 | 0.29 | 0.54 | 0.61 | 0.44 |
| 7. CH.3 | 0.88 | 0.42 | 0.37 | 0.45 | 0.29 |
| 8. CH.4 | 0.90 | 0.26 | 0.41 | 0.52 | 0.35 |
| 9. CH.5 | 0.89 | 0.33 | 0.35 | 0.46 | 0.30 |

• *l. 259: . . . five channels are all used -> . . . all five channels are used*

Reply: Thanks for the comment. It has been corrected.

• *l. 276: . . . focused on here.*

Reply: Thanks for the comment. It has been corrected.

• *l. 309: The PDs at . . . : Some text seems to be missing here.*

Reply: I am sorry for the mistake. The description has been revised.

• *l. 309: The PDs do not share the distribution characteristics of the low TBs. That would be in contradiction of PDs occurring mostly in warmer ice clouds and outflow regions.*

Reply: Thanks for the comment. The description has been revised. The changes are as:

"The distribution characteristics of PDs at 150 GHz ($TB_V$-$TB_H$) are similar to the structure of the tropical cyclone, but significant PDs occur mainly in the warm ice clouds at approximately 200-250 K."

• *l. 379: Refine conclusions to assess effects of a single 150 GHz channels as well as both 150 Ghz in terms of instantaneous estimates as well as annual means.*

Reply: Thanks for the suggestion. The changes are as:

"Since 2C-ICE was used to train the networks, MWHS IWP is certainly approaching the 2C-ICE and similar to the IWP maps in Duncan and Eriksson (2018). There is no significant difference between the results of the three MWHS channel combinations on the map, but the IWP result using only the 183 GHz channels is lower at middle latitudes than the IWP results with the addition of the 150 GHz channels."

[Figure]

Figure 11. Global mean IWP maps for 2015 from MODIS, 2C-ICE, ERA5 and different channel combinations from MWHS. 2C-ICE is gridded on a 5° grid, while the other products are gridded on a 1° grid.

• *Fig. 10: Please change the comparison to 2C-ICE to a line plot include this as an additional figure. I suggest you also add results from a retrieval using only channels around 183 as well as a retrieval using Ch. 2 - 5.*

Reply: Thanks for the suggestion. The changes are as:

"For tropical cyclone retrieval, the addition of the 150 GHz channel does not have a significant impact on the accuracy. The RMSE and CC of the three retrievals are similar. Although there are differences between MAPE and BIAS, the differences are not significant."

Table 6. Errors of the tropical cyclone retrieval

|          | RMSE (g m$^{-2}$) | MAPE (%) | BIAS (g m$^{-2}$) | CC   |
| -------- | ----------------- | -------- | ----------------- | ---- |
| CH. 1-5  | 1191.3            | 77.69    | 82.07             | 0.73 |
| CH. 2-5  | 1197.3            | 82.98    | 18.22             | 0.72 |

| | | | | |
|---|---|---|---|---|
| CH. 3-5 | 1174.1 | 79.71 | –113.67 | 0.73 |

[Figure]

Figure 10. IWP comparison of MWHS and 2C-ICE at the tropical cyclone Bansi.

• *Fig. 12: Here I suggest you also include results from a retrieval using only the channels around 183 as well as a retrieval using Ch. 2 - 5.*

Reply: Thanks for the suggestion. The changes are as follows:

"The IWP from MWHS is generally close to 2C-ICE, and the result without the 150 GHz channel is significantly lower than 2C-ICE between 30°S - 60°S in the Northern Hemisphere and 20°N - 60°N in the Southern Hemisphere. There is an improvement after adding the 150 GHz channel (little difference between using 1 or 2 150 GHz channels), and the IWP in the Northern Hemisphere is basically the same as the 2C-ICE, while it is still lower in the Southern Hemisphere."

[Figure]

Figure 12. Zonal means of IWP for 2015 from MODIS, 2C-ICE, ERA5 and different channel combinations from MWHS. 2C-ICE is gridded on a 5° grid, while the other products are gridded on a 1° grid.

• *l. 361: If you mention quantile regression neural networks, please cite Pfreundschuh et al. 2018.*

Reply: I am sorry for the mistake. The reference has been added.

• *l. 376: See comment above on impact of Ch. 4*

Reply: Thanks for the comment. It has been corrected.

• *l. 379: Refine conclusions to assess effects of a single 150 GHz channels as well as both 150 Ghz in terms of instantaneous estimates as well as annual means.*

Reply: Thanks for the comment. It has been corrected.

**Reviewer 2**

We would like to sincerely thank the reviewers for their professional comments and helpful suggestions. We believe they help us to improve the manuscript significantly and provide many useful ideas to our work. This manuscript has been revised by native English editors. We have revised the manuscript according to the reviewer's comments and answered the reviewer's question point by point below.

Reviewer comments are in italic blue and our responses in black, the manuscript changes are in red.

**Reply to comments**

*1. More statistical plots of the retrieval database are suggested to provide. An interesting figure should be the two-dimensional histogram between IWP and TB for different channels. Since the IWP-TB relationship has been revealed by both forward model simulations and remote sensing measurements, these plots could help to verify if the collocation database captures the right statistics. Also, the correlations between TB/PD and the scanning angle in the database are worth to be investigated. The results in Table 3 show that adding scanning angle as auxiliary information cannot improve the retrieval accuracies. Since TB and PD are both significantly influenced by the incidence angle, this geometry viewing information should be able to better constrain IWP in the state space. Additional plots in this aspect could make the discussion clearer.*

Reply: Thank you very much for the comments. We agree with the criticism and two statistical figures have been added. The changes are as:

"The statistical information of TB and IWP for different channels (CH.2 – CH.5) in the collocation database is given in Fig. 4. The TBs for CH.3 and CH.4 were mainly concentrated at approximately 250 K, indicating small sensitivity to ice clouds. For CH.2 and CH.5, the TBs had a larger range of variation, which is due to the larger contribution of near-surface information to the "window" channels. However, it can be seen that in the presence of ice clouds (IWP >100 g m-2), the surface information is blocked by clouds, making the TB range significantly smaller as the IWP becomes larger. The statistical relationship between the 150 GHz TB

and IWP at different scan angles is given in Fig. 5. It can be found that there is a significant decrease in the measured TB with increasing IWP for large scan angles. As the scan angle decreases, especially in the case of nadir observations, there are many low TBs appearing in clear-sky scenes because nadir observations have a very large number of collocation scenes in the polar regions (see Fig. 2b), where the surfaces lower the measured TB. In contrast, collocation scenes with large scan angles are mainly located in the tropics, which makes the TB-IWP relationship very significant."

[Figure]

Figure 4. Statistical information of TB and IWP for different channels.

[Figure]

Figure 5. Statistical information of IWP and 150 GHz TB for different scan angles.

The incident angle and observation geometry information should be able to better constrain the IWP in state space, which is why Holl et al. (2014) used local azimuth and zenith angles in training. Here we just used the scan angle of the satellite, which does not fully represent the geometric information, so it is of little help for the inversion, but combining it with the latitude information can be of better use.

*2. I suggest investigating the sparsity of the retrieval database in the measurement space when conducting retrievals, at least for the case study in section 4.3.1. You can plot the TB of each channel against that of all the other channels for the database and measurements, respectively, as the fig.2 in Brath et al., 2018 shows. Alternatively, you can calculate the $\chi^2=(y_{obs}-y_{db})^T*S_y^{-1}*(y_{obs}-y_{db})$ and check the $\chi^2$ values for the several best cases. If the training database does not well cover the full range of measurements, the NN retrievals cannot be expected to produce accurate results. This investigation could help to find the limitation of the retrieval database and better explain the retrieval results.*

Reply: Thanks for the comments. The changes are as:

"Figure 9 gives the TB of each channel against that of all the other channels for the training dataset (blue) and the validation dataset (red). Overall, the training dataset covers the full range of the validation dataset, which means that the neural network is well representative."

[Figure]

Figure 9. Measurements comparison from different channels of train data set (blue) and valid data set (red).

*3. Since very similar work has already been done by Holl et al., 2014, you should explicitly discuss whether your results are consistent with the results in Holl et al., 2014. Either direct or relative comparison is fine, and more statements regarding the improvements and extra findings of this work should be added. By the way, could you comment on why the biases in Tables 3-5 are always negative? Shouldn't the retrievals be unbiased?*

Reply: Thanks for the comments. The changes are as follows:

"In terms of the retrieval using the neural network, the results of this paper are basically consistent with Holl et al., (2014). The error between the retrieval results and 2C-ICE is approximately 100%. The latitude and ocean/land mask are important auxiliary information for DNN retrieval. Holl et al., (2014) used angle information that contains geometric observations of the local zenith and azimuth and showed a significant improvement. However, the results in Table 3 show that the scan angle is of limited help for retrieval, due to the fact that the scan angle is not fully representative of the geometry of the observed radiance, and it works better when used in conjunction with the latitude and land/sea mask."

The main improvements and findings of this work is the use of polarized 150 GHz channels. We have further supplemented the relevant channel combination comparison (with only using 183 GHz channels) in accordance with the comments of Reviewer 1 (see Figs 13, 14, 15). Related statements have been mentioned in the discussion and conclusion.

In the NN training, we chose the logarithm of the IWP as the training target and used the absolute percentage error as one of the main references for selecting the model, which make the model results underestimating the

true value, as mentioned previously by Reviewer 1. In addition, due to the difference in sensitivity of MWHS and CloudSat to ice cloud particles, it will lead to biased results.

---

## Author Response (AR3)

**Response to Reviewers**

We would like to sincerely thank the reviewers for their professional comments and helpful suggestions. We believe they help us to improve the manuscript significantly and provide many useful ideas to our work. We have revised the manuscript according to the reviewer's comments and answered the reviewer's question point by point below.

Reviewer comments are in italic blue and our responses in black, the manuscript changes are in red.

**Reviewer 1**

**Reply to comments**

- *l. 11: . . . in its retrieval.*

Reply: Thanks, revised.

- *l. 19: . . . of inputs of auxiliary variables and . . .*

Reply: Thanks, revised.

- *l. 24: ... in the midlatitudes ...*

Reply: Thanks, revised.

- *l. 91: The scanning is across the orbit, isn't it?*

Reply: Yes, the scanning of MWHS is across the orbit.

- *l. 241 . . . the training data (or collocations) are well representative.*

Reply: Thanks, revised.

- *l. 290: "So is the ocean/land mask information.". I guess you want to express that the land/ocean mask helps the retrieval distinguish hydrometeor scattering from the effect of the surface. However, that's not how I understand this sentence. Please consider reformulating it.*

Reply: Thanks. The sentence has been revised.

- *l. 306: I do not see the connection to Fig. 7. Do you mean Fig. 10?*

Reply: Thanks. The sentence has been deleted.

- *This is really a detail, but please consider using vector graphics for all line plots.*

Reply: Thanks. The vector graphics will be used and submitted separately in the final stage.

- *Fig. 7: Please add y-axis labels to all plots or at least the first plot of every row.*

Reply: Thanks. Y-axis labels have been added.

**Reviewer 2**

**Reply to comments**

1. *The biggest issue could be found in the IWP-TB relationship in figure 4. Since the topic of this study is to perform the global IWP retrievals, the NN training database is required to cover the entire possibilities in the measurement space. I did quick forward model simulations using the mid-latitude atmosphere/cloud profiles, and the IWP-TB relationship I get is shown below. The Comparison of these two figures indicates that the collocation dataset only captures a very small fraction of the possible TB range, especially when the IWP is over 100 g/m2. Similar conclusions can be drawn by comparing figure 4 and the MWHS measurement of the tropical cyclone in figure 12. For instance, the lowest TB of 183+-1GHz channel in the center of the cyclone reaches 180K, but the smallest TB value of the same channel in the collocation database is around 230K. The NN is impossible to handle such a level of extrapolations. The global TB*

*measurements used in section 4.3.2 are not given, and I believe there must be considerable amounts of TB measurements that are out of the coverage of the collocation database.*

[Figure]

*The IWP-TB relationship obtained from forward model simulations using mid-latitude atmosphere/cloud profiles.*

Reply: We are very sorry that our plots caused confusion. In fact, the collocation database does cover the TB range mentioned by the reviewer. In the revised version, we have updated Figs. 4 and 5 (essentially the corresponding colorbar) for a better representation of all TB and IWP cases considered in the NN training. It can be seen that the grey area corresponds to the cases with a smaller number of samples.

[Figure]

Figure 4. Statistical information of TB and IWP for different channels.

[Figure]

Figure 5. Statistical information of IWP and 150 GHz TB for different scan angles.

*2.    Figure 9 is not what I asked when I suggested investigating the sparsity of the measurement space in the last round of review. The training dataset and the validation dataset are both split from the collocation dataset, and there is no doubt they share the same statistics. What I intend to see is the comparison of TB in the training/validation database versus the cyclone and the global TB measurements applied in section 4.3. As discussed above, figure 4 shows that the collocation database is far from fully covering the TB space, and therefore we cannot expect the NN to produce sensible retrieval results.*

Reply: Thanks for your suggestion. Figure 9 has been updated using the MWHS dataset in 2015 (blue) and the collocation dataset (red). Due to a large amount of MWHS data, $10^6$ measurements were randomly selected for each month, i.e., a total of $1.2 \cdot 10^7$ measurements. We believe that the collocation dataset already covers the most range of the TB space and should be sufficiently representative in NN training. Nevertheless, there is no doubt that the collocation dataset is not possible to fully cover the measurement space due to the different orbit and scan methods.

[Figure]

Figure 9. Measurement comparison from different channels of MWHS measurements in 2015 (blue) and collocation dataset discussed above (red).

*3.    Another critical issue is the retrieval experiment in figure 11, which tests the NN retrieval accuracies by comparing the retrieved parameters with the reference IWP using a testing database. The testing dataset is obtained from the same collocation finding procedure as the training/validation dataset but over a different time. Although the testing dataset is not used in the training, a well-established NN is capable to produce very accurate results since the testing and training/validation datasets have very similar statistics. However, figure 11 shows the correlations between the retrieved IWP and the reference are terrible when IWP is smaller than 1 kg/m2. The authors say this "may be due to the lack of sensitivity of the MWHS to thin ice clouds" (line*

*319), but even figure 4 shows that the MWHS channels are sensitive to the IWP when it is over 100 g/m2. Figure 8 in Holl et al., 2014 conducted an identical testing experiment, and the NN results they got are consistent with the 2C-ICE along the whole range, which is in line with expectations. The training/validation dataset is undoubtedly one contributing factor to the poor performance, but we cannot eliminate the possibility that the NN is not appropriately implemented.*

Reply: Thanks for your comment. We believe that the result in Fig. 11 is normally interpretable and there is no problem with the collocation database or NN implementation. Although the MWHS channels are sensitive to the IWP when it is over 100 g/m$^2$, the ΔTB is considered to be small. Referring to Fig. 6 in Holl et al., 2014, IWP at 100 g/m$^2$ is where the microwave retrieval error is largest. Figure 8 in Holl et al., 2014 used the TIR channels to improve the performance in this IWP range, since the retrieval using the TIR channels performs better when IWP less than 1000 g/m$^2$ (see Fig. 5 in Holl et al., 2014). It should be noted that Fig. 8 in Holl et al., 2014 does not yet consider the cloud filter, whereas Figure 11 in our manuscript is the IWP retrieval after cloud filtering. Since the cloud filtering only using microwave channels does not perform satisfactorily compared to that using IR channels (used in Holl et al., 2014), the final retrieval result does not perform well at IWP of 100 g/m$^2$. However, the correlation of retrieval results seems better without considering the cloud filtering, see figure below.

[Figure]

Comparison between 2C-ICE and MWHS IWP without cloud filtering

[Figure]

Figure 5 in Holl et al., 2014                    Figure 6 in Holl et al., 2014

---

## Author Response (AR4)

**Response to Reviewers**

We would like to sincerely thank the reviewers for their professional comments and helpful suggestions. We believe they help us to improve the manuscript significantly and provide many useful ideas to our work. We have revised the manuscript according to the reviewer's comments and answered the reviewer's question point by point below.

Reviewer comments are in italic blue and our responses in black, the manuscript changes are in red.

**Reviewer 1**

**Reply to comments**

- *l. 213: "... to close the reference data.". I think this part of the sentence is incorrect and suggest to remove it.*

Reply: Thanks, revised.

**Reviewer 2**

**Reply to comments**

*General comments:*

1. *It is good to see the collocation database has broader coverage in fig.4, even though the samples in most areas are still limited. As for the retrievals of thin ice clouds in the testing experiments, I still feel uncomfortable about the NN results. Figure 11 and the first plot of fig. 13 suggest that the capabilities of IWP retrievals using MWHS measurements are around 1 kg m-2, and the retrieved IWP below this threshold will be significantly biased even when we have idealized prior knowledge. Both conclusions are not in agreement with previous studies.*

Reply: Thanks for your comment. I think the conclusions are consistent in general, the main inconsistency is due to the cloud filtering network, and the performance of microwave channels is limited. The IWP threshold of 1 kg m-2 is also caused by cloud filtering, it is not obvious in Fig. 9 (IWP retrieval without cloud filtering). Further research will be focused on this issue.

2. *I think it is fine to publish this manuscript to present the latest algorithm development with FY MWHS measurements for potential ice cloud products in the future. Considering that the collocation database in this study only contains one-year data and the number of cloudy samples is relatively small (<1e5), I suggest that the authors should keep improving the database in the future research since it is one of the most critical elements in the NN algorithm.*

Reply: Thanks for your suggestion. The collocation database is quite important in NN retrieval and the result is limited due to the one-year samples. More years of collocation data will be included in the future study.

*Technical issues:*

3. *l. 43, 52: Instruments like AMSU and GMI are referenced without introducing what they mean.*

Reply: Thanks for your comment. These instruments such as AMSU are referenced since they have similar channels to FY-3B/MWHS (but these sensors are without polarization channels) and have been used to measure IWP. This introduction has been added. GMI is referenced to demonstrate that the polarization differences are associated with ice clouds and this is described in the same paragraph (l. 53-57).

*4. l. 86: Descriptions like "subsequent" and "in the end" need to be more specific.*

Reply: Thanks for your comment. The descriptions have been revised as:

*"The IWP retrieval results and analysis are discussed in the Sect. 4.1 and Sect. 4.2. The network application on tropical cyclones and the global mean map are shown in Sect. 4.3, with conclusions in Sect. 5."*

*5. l. 139, 154, 166, 169: number needs to be written in the right format. For example, 1207731 should be 1 207 731.*

Reply: Thanks, revised.

*6. Fig. 2: the colorbar of panel(b) needs to be added.*

Reply: Thanks, revised.

*7. l. 197: Fig. 6 should be Figure 6*

Reply: Thanks, revised.

*8. l. 239-242: This figure does not fit the context here and it should be moved to section 4.3.2.*

Reply: Thanks, revised.

*9. Fig. 9: Channels for the y-axis need to be indicated.*

Reply: Thanks, revised.

*10. l. 321: use g m-2 instead to make the unit consistent.*

Reply: Thanks, revised.

*11. l. 332, 334, fig.12: change channels like 183-7 to 183\pm7 for consistency with fig. 4.*

Reply: Thanks, revised.